# Concurrence of *FGFR1* mutations modulates oncogenesis in glioneuronal tumors

Jacopo Boni[1,2], Míriam Fernández-González [ID][1,3], HyeRim Han [ID][1,3], Carla Roca[1,3], Cassandra J Wong[4], Cristina Rioja [ID][1], Clara Nogué [ID][1,3], Leticia Manen-Freixa [ID][1,5], Jonathan Boulais [ID][6], Endika Torres-Urtizberea [ID][7], Antonio Gomez [ID][8], Martin Hasselblatt [ID][9], Roger Estrada-Tejedor[7], Albert A Antolin[1,8,10], Islam E Elkholi [ID][6], Nada Jabado [ID][11], Jean-François Côté [ID][6,12,13], Anne-Claude Gingras [ID][4,14] & Barbara Rivera [ID][1,15,16 ✉]

## Abstract

*FGFR1* genetic alterations are associated with brain malignancies, including *FGFR1* mutations in familial and sporadic cases of low-grade glioneuronal tumors, suggesting intrinsic mechanisms of selective pressure toward *FGFR1* multiple events arising in the context of a quiet genome. To decipher the molecular mechanisms triggered by multiple concurrent *FGFR1* mutations, we have mapped the proximal interactome of wild-type, single- and double-mutant FGFR1 proteins through a BioID-MS approach. Our data reveal novel oncogenic functionality for the two hotspot mutations N546K and K656E, linked to evasion of lysosomal degradation. Further, we identified a modulatory tumor-suppressive role for the susceptibility variant R661P, which hampers the oncogenic potential of both hotspot N546K and K656E mutations by rescuing receptor degradation and reducing N546K affinity for the downstream effector PLCγ. Introducing the R661P missense variant was sufficient to abolish self-renewal capacity of oligodendroglioma cells and downregulate genes involved in neurodevelopment and neuroglial cell fate decisions, both aspects overcome in the double mutants. This study sheds light on contextual oncogenic effects associated with *FGFR1* alterations and their recurrence in low-mutation burden and therapy naive tumors.

**Keywords** FGFR1; Multiple Mutations; Glioneuronal Tumors; Cell Models; Modulatory Mechanisms
**Subject Categories** Cancer; Genetics, Gene Therapy & Genetic Disease; Neuroscience

## Introduction

Fibroblast growth factor receptors (FGFRs) are a highly conserved family of receptor tyrosine kinases (RTKs) playing fundamental functions during organ development and tissue homeostasis (Xie et al, 2020). Hence, genetic alterations in FGFR genes have been associated with a broad spectrum of human diseases. Constitutional pathogenic variants in *FGFR1* have been linked to the etiology of several developmental disorders, which comprise Pfeiffer syndrome, osteoglophonic dysplasia, Kallmann syndrome, and Hartsfield syndrome, among others (Dhamija and Babovic-Vuksanovic, 1993–2025; Jarzabek et al, 2012; Vogels and Fryns, 2006; White et al, 2005). On the other hand, somatic alterations such as *FGFR1* gene amplification, duplication of the tyrosine kinase domain, gene fusions and hotspot single-nucleotide variants (SNV) have been found enriched in many types of solid tumors, including lung squamous cell carcinoma, urothelial carcinoma and glioma (Helsten et al, 2016).

Our group and others have described *FGFR1* genetic variants linked to the etiology of hereditary and sporadic forms of early-onset, low-grade glioneuronal tumors (LGGNTs). Specifically, duplication of the kinase domain, gene fusions and point mutations were identified as driver events in pilocytic astrocytoma, rosette-forming glioneuronal tumors (RGNTs) and other LGGNT subtypes (Engelhardt et al, 2022; Jones et al, 2013; Qaddoumi et al, 2016; Rivera et al, 2016; Sievers et al, 2019; Zhang et al, 2013). In

[1]Bellvitge Biomedical Research Institute (IDIBELL), Avinguda de la Granvia de l'Hospitalet 199, 08908 L'hospitalet de Llobregat, Barcelona, Spain. [2]Centro de Investigación Biomédica en Red de Cáncer (CIBERONC), Madrid, Spain. [3]Universitat de Barcelona (UB), Gran Via de les Corts Catalanes, 585, Barcelona 08007, Spain. [4]Lunenfeld-Tanenbaum Research Institute, Mount Sinai Hospital, Sinai Health System, Toronto, ON, Canada. [5]Center for Cancer Drug Discovery, The Division of Cancer Therapeutics, The Institute of Cancer Research, London, UK. [6]Institut de Recherches Cliniques de Montréal (IRCM), Montreal, QC H2W 1R7, Canada. [7]Grup de Química Farmacèutica, IQS School of Engineering, Universitat Ramon Llull, Via Augusta 390, Barcelona E-08017, Spain. [8]Department of Biosciences, Faculty of Sciences and Technology (FCT), University of Vic - Central University of Catalonia (UVic-UCC), Vic, Barcelona, Catalonia 08500, Spain. [9]Institute of Neuropathology, University Hospital Münster, Münster 48149, Germany. [10]Program Against Cancer Therapeutic Resistance (ProCURE), Catalan Institute of Oncology (ICO)/IDIBELL, L'Hospitalet de Llobregat, Barcelona, Spain. [11]Department of Human Genetics, McGill University, Montreal, QC, Canada. [12]Department of Medicine, Université de Montreal, Montreal, QC, Canada. [13]Department of Anatomy and Cell Biology, McGill University, Montreal, QC, Canada. [14]Department of Molecular Genetics, University of Toronto, Toronto, ON, Canada. [15]Lady Davis Institute for Medical Research, Segal Cancer Centre, Jewish General Hospital, 3755 Chemin de la Côte Sainte-Catherine, Montreal, QC H3T 1E2, Canada. [16]Gerald Bronfman Department of Oncology, McGill University, Montreal, QC H4A 3T2, Canada. ✉E-mail: brivera@idibell.cat

particular, a high proportion of pediatric epileptogenic LGGNTs, defined as dysembryoplastic neuroepithelial tumors (DNETs), harbor oncogenic *FGFR1* alterations (Qaddoumi et al, 2016; Rivera et al, 2016). Histologically, these tumors are characterized by both neuronal and oligodendroglia-like elements (Daumas-Duport, 1993; Louis et al, 2021). Our study identified a hereditary form of DNETs with a novel *FGFR1* missense variant (R661P) mapping in the tyrosine kinase domain. Notably, additional hotspot *FGFR1* somatic mutations (either N546K or K656E) were detected *in cis* to the R661P variant in the tumors of carrier individuals (Rivera et al, 2016). Although double mutations in driver genes are rare events, the pattern of multiple, co-occurring single-point mutations *in cis* (hereafter referred as "multiple") in *FGFR1* gene was validated in sporadic cases. These findings point to mechanisms of selective pressure that promote the accumulation and positive selection of multiple mutational events in *FGFR1* during glioneuronal tumor formation. This occurrence of intragenic multiple mutations in low-grade, low mutation-burden and predominantly therapy-naive tumors, represents a striking and unexplained phenomenon.

*FGFR1* N546K and K656E oncogenic variants have been reported in postzygotic mosaicism as the genetic cause of a rare neurocutaneous condition known as Encephalocraniocutaneous lipomatosis (ECCL) (Bennett et al, 2016); however neither variant has so far been reported in the germline, suggesting embryonic lethality. ECCL has been classified as one of the RASopathies, a group of developmental disorders characterized by germline or mosaic mutations in genes that lead to constitutive activation of the RAS/MAPK pathway. Additional *FGFR1* missense mutations *in cis* have been reported in low-grade gliomas (LGGs) developed by ECCL individuals classified as midline pilocytic astrocytomas (Bennett et al, 2016; Valera et al, 2018), providing additional evidence for a driver role of multiple *FGFR1* mutations in LGGs/LGGNTs.

Molecularly, the two hotspot mutations have been linked to increased basal activation and ectopic expression has been shown to induce transformation and hyperproliferative phenotypes in mammalian cell lines (Cimmino et al, 2022; Hart et al, 2000; Lew et al, 2009; Yoon et al, 2004), indicating an acquired oncogenic potential by the mutated receptor. Despite this data, the precise pro-tumorigenic molecular mechanisms in glial cells remain elusive. Moreover, to date no research work has investigated how this tumorigenic activity is affected by additional *FGFR1* mutations *in cis* and why they are positively selected in LGGNTs (mostly low-mutation burden and therapy-naive tumors). In the present study, we employ complementary models and experimental strategies, including a Proximity-dependent Biotin Identification (BioID)-based proximal interactome profiling and CRISPR-engineered glioma cell lines, to investigate how secondary hits in the oncogene *FGFR1* modulate cellular effects and oncogenic drive mediated by hotspot mutations.

Overall, our findings provide mechanistic evidence on the driver effects exerted by multiple mutations in *FGFR1* gene and shed new light on tumorigenic mechanisms during early-onset brain tumor formation.

# Results

## Hotspots and multiple *FGFR1* mutations are recurrent in brain tumors

We queried the GENIE database (de Bruijn et al, 2023) (https://genie.cbioportal.org/) to define the spectrum of tumors harboring at least one of the two hotspot variants N546K and K656E. These two amino acid changes represent the most frequent event among all the missense variants identified for the residues Ans546 (213/244) and Lys656 (121/137) (Appendix Table S1). High specificity for brain tumors was confirmed for both variants (Fig. 1A; Table 1; $\chi^2$ test, $P < 0.001$). Among these cases (N546K or K656E mutated), we reviewed tumors with multiple hits in *FGFR1* and found that they were exclusive to brain tumor types, with the hotspot K656E being the one more frequently accompanied by additional *FGFR1* hits (Fig. 1A; Table 2). Secondary *FGFR1* variants ("secondary" hereafter referred as any *FGFR1* missense variant other than N546K or K656E identified in samples with multiple *FGFR1* variants) have been specified in Appendix Fig. S1. By examining tumor grade of hotspot-positive brain tumor types, we found that multiple *FGFR1* alterations are found in similar rates in both low-grade (grades 1–2) and high-grade (grades 3–4) cases, with a higher number of double-mutant cases identified in low-grade types (Fig. 1B; Table 3). Regarding the timing of occurrence and clonal selection of *FGFR1* multiple mutational events, allele frequencies do not provide enough information on whether "secondary" variants appear before or after the occurrence of the oncogenic hit. Nevertheless, we can infer hypothesis on the timing from patients with genetic disorders caused by germline and mosaic forms of *FGFR1* variants, which indicate that both scenarios ("secondary" hits appearing first as in familial DNETs, or hotspots appearing first as in ECCL) are possible (Fig. 1C).

## Proximal interactome of wild-type and mutant FGFR1 receptors

To investigate how different mutations rewire the network of molecular interactions engaged by the FGFR1 receptor, we employed a systematic interactome-profiling approach using the BioID strategy coupled to mass spectrometry (MS). The approach is based on the biotinylation of proteins within a 10 nm radius of the bait mediated by the mutant enzyme BirA* in the presence of biotin (Roux et al, 2012) and allows the identification of interactions hard to capture with standard co-immunoprecipitation strategies. For the purpose of this study, we chose to focus on the two hotspots N546K and K656E, the germline variant R661P, previously identified by our research team (Rivera et al, 2016), and both combinations *in cis* (N546K/R661P and K656E/R661P) as models of "hotspot + secondary" double mutants (Fig. 2A,B). Toward this aim, the Flp-In T-REx HEK293 system has been used to generate inducible, stable cell lines expressing the coding sequences of WT and mutant FGFR1 receptors, fused to the mutant enzyme BirA* and Flag (Fig. 2B). The expression and activation (phosphorylation of activation loop Tyrosine residues 653/654) of the six FGFR1-BirA*-Flag fusion proteins were validated by western blot (Fig. 2C), demonstrating proper kinase function of the BirA*-fused C-terminal domain.

The rationale of our experimental design was based on a BirA* enzyme fused to the C-terminus of the FGFR1 protein, providing labelling of intracellular factors in the proximity of the receptor, which have been purified through streptavidin affinity pull-down and identified through mass-spectrometry (Fig. 2D). Biotinylation of proximal interactors was confirmed for all the baits by western blotting (Appendix Fig. S2). Despite the observed differences in protein amounts among conditions (Fig. 2C, discussed in the next section) we did not obtain any bias in the outcome of our

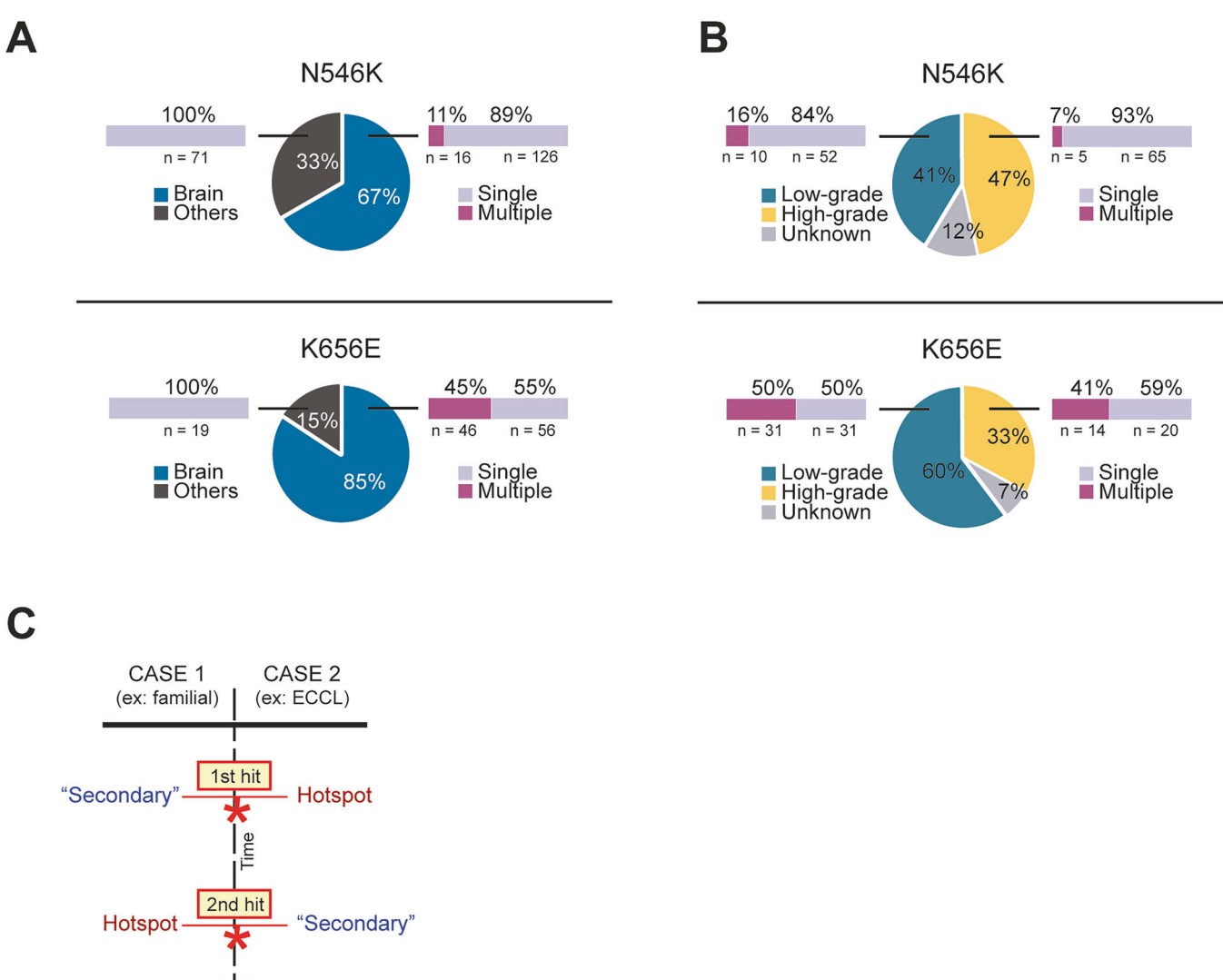

**Figure 1. Genotype-phenotype analysis of *FGFR1* hotspots N546K and K656E, alone or with secondary hits.**

(A) Distribution of N546K ($n = 213$) and K656E ($n = 121$) tumors comparing Brain vs no-brain, or "Others" tumor types. Numbers and percentages of cases presenting only the hotspot (single) vs hotspot + other hits in *FGFR1* (multiple) are indicated for each group. Both categories (single and multiple) are significantly associated with brain tumor types (Chi-squared test, *P* value < 0.0001). (B) Distribution of N546K and K656E brain tumors comparing low-grade (grade 1–2) vs high-grade (grade 3–4) tumor types. Numbers and percentages of cases presenting only the hotspot (single) vs hotspot + other hits in *FGFR1* (multiple) are indicated for each group. (C) Timeline of acquisition of hotspot (either N546K or K656E) and "secondary" hits. The two possible scenarios with respective examples (familial case and ECCL patients) are indicated. ECCL encephalocraniocutaneous lipomatosis.

interactome profiling, as indicated by comparable total numbers of identified unique preys (Fig. 2E; Dataset EV1). Our BioID capture was able to recall FGFR1 (WT) interactors reported in the BioGRID database (Oughtred et al, 2021), including proximal interactors ($n = 81$) identified in a recent comprehensive study using human RTKs as baits expressed through the same cell model (Flp-In T-REx HEK293) (Salokas et al, 2022) (Fig. 2E; Dataset EV1). This data supports the reliability of our experimental procedures and, on the other hand, highlights potential novel, so far unreported FGFR1 protein partners. Among the recalled preys, we also retrieved many well established FGFR1 bona fide interactors, including other members of the FGFR family (FGFR2 and FGFR3), modulators as protein tyrosine phosphatase non-receptor type 1, 11 (PTPN1,

PTPN11) and downstream signaling effectors as fibroblast receptor substrate 2 (FRS2), phospholipase C gamma 1 (PLCγ/PLCG1) and signal transducer and activator of transcription 3 (STAT3) (Fig. 2F). The composition of each interactome was compared, showing both unique and shared preys among FGFR1 mutants and with the WT protein (Fig. 2G). The most abundant group ($n = 167$) was formed by proteins detected in every condition, confirming that differences in protein expression had not a strong impact in the BioID-MS and highlighting a core of functionality not affected by the mutations considered in this study (Fig. 2G). To further elucidate similarities and divergences among the WT and mutant proteins, we performed an unsupervised hierarchical clustering analysis. The germline R661P clustered differently (and closer to the WT

**Table 1. Number of cases in the GENIE database harboring the N546K or K656E mutation.**

| Tumor tissue | N of cases |
|---|---|
| **N546K** | |
| Brain | 142 |
| Others | 71 |
| Total | 213 |
| **K656E** | |
| Brain | 102 |
| Others | 19 |
| Total | 121 |

Samples are categorized based on their classification as brain tumors or other tumor types.

**Table 2. Number of patients with *FGFR1*-mutated brain tumors in the GENIE database.**

| FGFR1 hits | N of cases |
|---|---|
| **N546K** | |
| Single (hotspot only) | 126 |
| Double/multiple | 16 |
| Total | 142 |
| **K656E** | |
| Single (hotspot only) | 56 |
| Double/multiple | 46 |
| Total | 102 |

Samples are distributed based on whether they present only one of the hotspots mutations N546K or K656E ("Single") in *FGFR1* or they display additional missense mutations in *FGFR1* ("Double/multiple"). Of note, among all the cases annotated as Double/multiple, only 3 patients harbored more than two mutations (all appearing with the hotspot K656E).

**Table 3. Number of *FGFR1*-mutated brain tumor samples in the GENIE cohort, categorized by tumor grade and divided in Single and Double/multiple (same as Table 2).**

| Tumor grade | FGFR1 hits | | |
|---|---|---|---|
| | Single (hotspot only) | Double/multiple | Total |
| **N546K** | | | |
| Low-grade | 52 | 10 | 62 |
| High-grade | 65 | 5 | 70 |
| Unknown | 17 | 1 | 18 |
| Total | 134 | 16 | 150 |
| **K656E** | | | |
| Low-grade | 31 | 31 | 62 |
| High-grade | 20 | 14 | 34 |
| Unknown | 5 | 2 | 7 |
| Total | 56 | 47 | 103 |

Low-grade: tumor types with grade 1 or 2; high-grade: tumor types with grade 3 or 4; unknown: tumor types that could not be assigned.

protein), compared to the hotspot mutants, both single and double (Fig. 2H). On the other hand, although they clustered similarly to their respective single mutants N546K and K656E, double mutants displayed specific interactors (Fig. 2H, Clusters 2, 5 and 6; Dataset EV1). This overview of the interactome already indicated the uniqueness of the R661P variant and potential modulatory effects that it may exert when combined with the oncogenic mutations.

## FGFR1 oncogenic mutants rapidly undergo autophosphorylation and are highly stable in human cells

Coherently with their activating nature, cells expressing the hotspot mutations N546K and K656E appeared with increased phosphorylated FGFR1 (p-FGFR1) levels and enhanced activation of downstream MAPK/ERK pathway, compared to the WT condition (Figs. 2C and EV1A). However, by looking at the total Flag-FGFR1 levels, we observed a similar trend, implying that the enhanced signal was not solely due to intrinsic phosphorylation, but also reflected increased protein accumulation (Fig. 2C). To further test this, total Flag-FGFR1 protein levels from multiple independent experiments were quantified, revealing a significant and high (4–6-fold change) increase in the levels of accumulation

of the oncogenic mutant proteins, both in single and double settings (Fig. 3A). On the contrary, when we assessed intrinsic autophosphorylation rates (phopsho-FGFR1 levels corrected against total Flag-FGFR1 levels) the differences between oncogenic mutants and WT protein were less pronounced and statistically non-significant, except for N546K (Fig. EV1B). Similar total and phosphorylated protein profiles were observed in cells previously serum-starved, indicating that, in our settings, differences among conditions do not depend on the presence of growth factors (Fig. EV1C). Notably, when we looked at specific interactors in BioID-MS for this group (Fig. 2H, Cluster 11), we found an enrichment in GO categories related to protein transport and vesicles trafficking, which might indicate altered regulation of receptor turnover (Fig. 3B), potentially affecting recycling and degradation rates (Miaczynska, 2013). Consistent with previous findings on N546K (Lew et al, 2009), HEK293 cells overexpressing an oncogenic mutant FGFR1 (N546K and K656E, but also double mutants N546K/R661P and K656E/R661P) displayed a transformed phenotype with changes in morphology toward a stem-like shape and higher tendency to detach from the plate (Fig. EV1D), large nuclei and reduced cytoplasmic volume (Fig. 3C). In addition, we observed different FGFR1 subcellular localization between WT and oncogenic protein-expressing cells. While WT and R661P mutant display classic receptor localization, mainly in the plasma membrane with little cytoplasmic involvement, oncogenic mutants accumulate at high levels in intracellular compartments (Fig. 3C), reinforcing the hypothesis of a dysregulated receptor transport for these conditions. Coherent with these changes in localization, analysis of groups of interactors identified exclusively for R661P and WT FGFR1 (Fig. 2H, Clusters 14–15; Dataset EV1) were found enriched in plasma membrane and cytoskeleton-associated proteins (Fig. EV1E).

Considering the homogenous expression ensured by the Flp-In system (transcript levels are represented in Fig. EV2A, with no condition exceeding a two-fold change variation), the increased levels observed at 24 h post-Tet induction are most likely due to

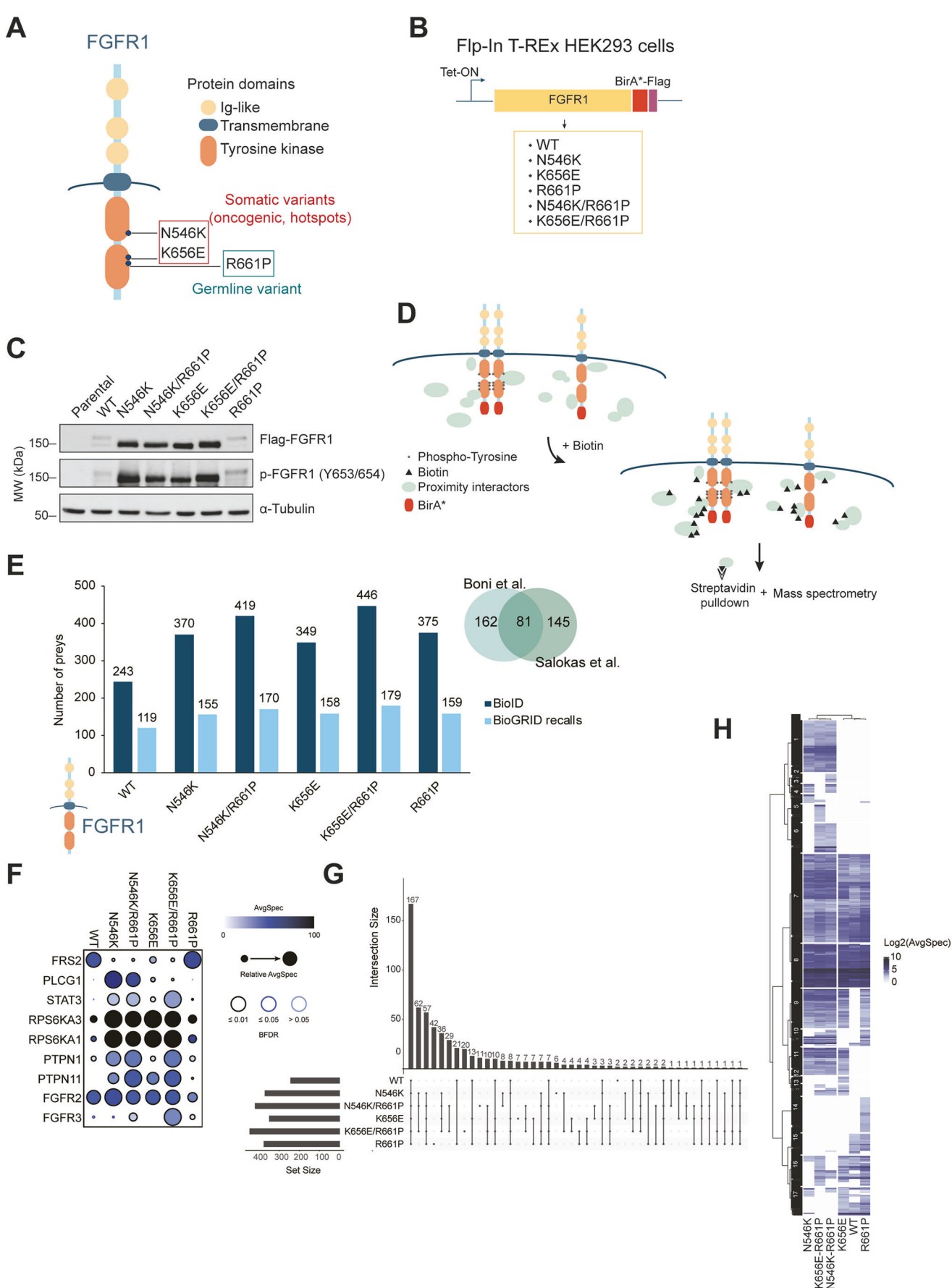

**Figure 2. Proximal interactome profiling (BioID-MS) of WT and mutant FGFR1 proteins.**

(A) Scheme of the FGFR1 receptor representing functional domains and SNVs investigated in the present study: the hotspots N546K, K656E and the germline variant R661P. (B) Stable inducible HEK293 (Flp-In T-REx) cell lines have been generated to express the coding sequence of WT and the 5 mutant FGFR1 fused mutant BirA* enzyme and Flag. (C) Western blot of the parental (empty-Flag) and the six Flp-In T-REx HEK293 cell lines used in the present study, showing expression and activation (phosphorylation of Tyr653/654) of FGFR1, after 24 h of Tet induction. (D) A schematic outline of the BioID-MS screen: in the presence of biotin the BirA* enzyme fused to both active (phosphorylated dimers) and inactive FGFR1 molecules catalyzes the biotinylation of proximal interactors, which are isolated through streptavidin-mediated pull-down and analyzed through MS. (E) Number of identified preys (after filtering) as well as BioGRID recalls for each bait are represented. Right: numbers of shared and non-shared interactors of WT FGFR1 between our list and the one published by (Salokas et al, 2022). (F) Dot plot highlighting FGFR1 bona fide interactors and their abundances (AvgSpec) among the different baits. (G) Upset plot showing numbers of proximal interactors among different combinations of intersections of the different baits. (H) Heatmap of an unsupervised hierarchical clustering analysis using bait abundances (Log2AvgSpec). 17 distinct clusters have been identified. AvgSpec average of spectral counts. Source data are available online for this figure.

post-translational mechanisms occurring during this time frame. To explore early events occurring after FGFR1 mutant synthesis and accumulation, we collected cell lysates at 8 h of induced expression. At 8 h, all mutants, including N546K and K656E single mutants, revealed a WT-like profile, with two distinct bands, highlighting similarities in the post-translational processes they engage at early stages (Fig. EV2B). However, expression levels of oncogenic mutants were already higher at 8 h compared to WT (Fig. EV2B), suggesting that part of the WT protein reservoir is already targeted for degradation during early stages of protein expression, through mechanisms that are evaded by oncomutant proteins. To better define dynamics of FGFR1 receptor accumulation and autophosphorylation, we performed time-course experiments for each condition, assessing total Flag-FGFR1 and p-FGFR1 levels at five different timepoints after Tet induction. The peak of accumulation for WT and R661P FGFR1 proteins is reached between 8 and 12 h, with levels decreasing from 12 to 24 h, likely indicating enhanced degradation rates as a cellular response to protein overexpression. On the contrary, oncogenic mutants maintain stable levels in this last time frame, reinforcing the hypothesis of escaped degradation (Fig. 3D). Notably, phosphorylation of oncogenic mutant receptor was also enhanced at 8 h, confirming that hyperphosphorylation is an early event driven by the amino acid changes N546K and K656E, with a stronger phenotype presented by the first one. These findings suggest that the two properties, hyperphosphorylation and high protein stability, might be mechanistically linked and synergically cooperate in mediating oncogenic phenotypes in tumor cells. To explore this hypothesis, we next treated the cells with the FGFR inhibitor (FGFRi) AZD4547 4 h after inducing FGFR1 expression and analyzed each condition at 24 h (Fig. EV2C). As shown in Fig. 3E, treatment with AZD4547 abolished WT and mutant FGFR1 autophosphorylation. Strikingly, while un-phosphorylated WT FGFR1 protein shows total levels comparable to control (vehicle), all oncogenic mutants "rescued" a WT-like profile (band pattern and intensity) when treated with the inhibitor (Fig. 3E), which reflects a dramatic decrease in oncogenic protein accumulation, compared to control (DMSO) condition. The consequences of this regression were also seen in the activation of the downstream MAPK/ERK signaling pathway (Fig. 3E), and at the phenotypic level, as oncogenic mutants re-attached to the plate upon treatment with FGFRi, reverting the transformed status (Fig. EV2D). This data indicates that enhanced tyrosine kinase activity and high protein stability, are linked and that hyperphosphorylation is required for the escape of degradation processes caused by N546K and K656E FGFR1 mutations.

## Secondary mutations partially rescue lysosomal degradation of oncogenic proteins

Functional analysis of our BioID interactome data, enriched in endosomes- and vesicle transport-related categories, together with the molecular evidence obtained through the experiments described so far, point towards dysregulated lysosomal processes. To assess whether, in normal conditions, WT FGFR1 expressed in the T-REx HEK293 cells is mainly targeted to degradation in lysosomes and whether K656E and N546K hotspot mutations impair lysosomal degradation, we treated the cells with Bafilomycin A1 (BafA1), a potent inhibitor of vacuolar H$^+$ ATPases and therefore of lysosomal function (Yoshimori et al, 1991). Upon BafA1 addition, WT and R661P FGFR1 displayed higher protein levels and an "oncogenic mutant-like" pattern with the lower molecular weight band that accumulates over the upper one (Fig. 4A), suggesting that, in normal conditions, i) this low-molecular weight protein form is the most efficiently targeted for degradation in lysosomes in WT settings, and ii) oncogenic mutant FGFR1 receptors escape lysosomal degradation, accumulating only the faster-resolving band (Fig. 4A). To further investigate how oncogenic mutants evade vesicular degradation pathways, we induced receptor internalization and lysosomal degradation by stimulating the FGFR1-expressing HEK293 cells with the FGFR ligand FGF2 (Fig. 4B). In parallel, to avoid accumulation of newly synthesized protein, we inhibited mRNA translation by treating the cells with cycloheximide (CHX) (Fig. 4B). While WT levels decreased over time, the oncogenic mutant K656E escaped degradation, maintaining stable protein levels (Fig. 4C,D). Notably, this degradation was partially rescued in the double mutant, with protein levels decreasing to 50% at 6 h (Fig. 4D), indicating a novel modulatory mechanism played by the R661P variant over the oncogenic mutant K656E. Supporting these results, upon treatment with Bafilomycin, degradation was prevented and similar levels of WT and K656E/R661P proteins (38 and 31%, respectively) were recovered at 6 h (Fig. 4E,F). The destabilization exerted by the double mutant K656E/R661P, compared to the single K656E mutation, was similarly confirmed for the N546K/R661P mutant, which partially rescued protein degradation relative to N546K (45% of starting protein levels) (Fig. EV3A,B). These results highlight similar mechanisms wielded by the R661P to enhance degradation of both oncogenic mutants. This modulatory effect played by the secondary mutation also affected downstream pathway activation, as ERK1/2 phosphorylation was also reduced at 6 h in double mutant-expressing cells (Figs. 4C and EV3A). To exclude the possibility that these effects were R661P variant-specific, we sought to validate

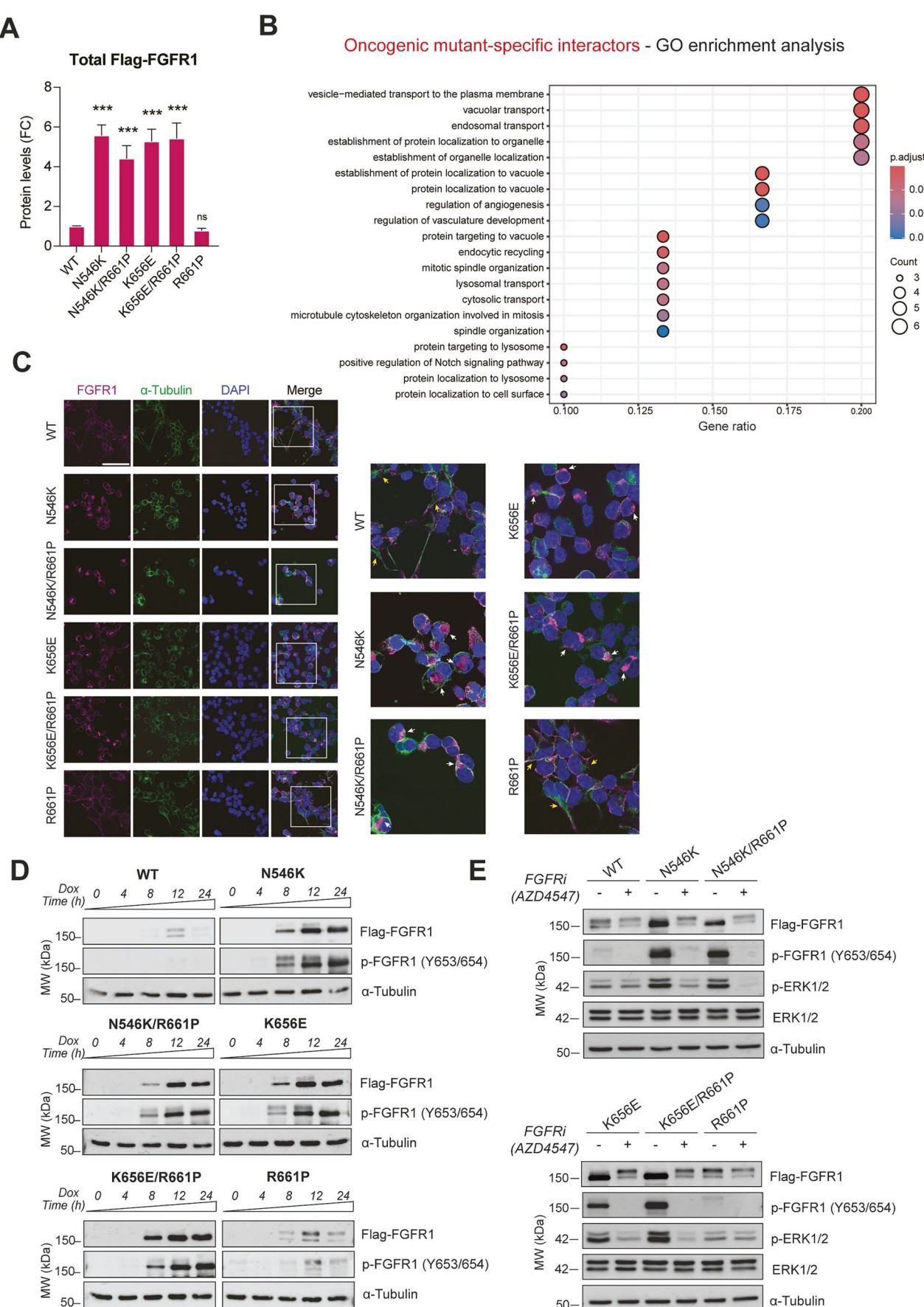

**Figure 3. N546K and K656E mutated receptors show increased phosphorylation and protein stability.**

(A) Bar plot of relative amounts (quantifications of western blots from n = 6 independent experiments) of total WT and mutant Flag-FGFR1 protein levels obtained from the six T-REx HEK293 cell lines, induced with doxycycline (24 h). Fold change (FC) values, obtained normalizing against WT levels, are indicated. Error bars represent standard errors of the mean ± SEM values. Significant comparisons against WT values are indicated (Kruskal–Wallis test); P values: ***P < 1e09. (B) Dot plot of GO enrichment analysis using the list of genes included in Cluster 11 of the heatmap in Fig. 2H (shared interactors among the 4 hotspot-expressing cell lines) as input list. Significantly enriched (adj. P value ≤ 0.05, Benjamini–Hochberg method) have been represented. The size and color of dots represent numbers of matched genes and adjusted P value (P.adjust), respectively. (C) Immunofluorescence staining of FGFR1 protein (pink) and α-tubulin (green) using the T-REx HEK293 FGFR1 cell lines, displaying differences in cellular localization of FGFR1 receptors. Right panels: zoom in (white squares in left panels). Arrows highlight different patterns of FGFR protein in the oncogenic mutants (white arrows indicating accumulation in intracellular compartments), compared with WT and R661P (yellow arrows indicating FGFR1 localization to the plasma membrane). Scale bar: 50 μm. (D) Time course of Flag-FGFR1 protein synthesis and phosphorylation showing differential accumulation rates and patterns among the different mutants. (E) Western blot assays revealing total and phosphorylated FGFR1 levels, alongside with total and phosphorylated ERK1/2 levels in response to the FGFR inhibitor (FGFRi) AZD4547 (10 nM), compared to control (DMSO) condition. GO = Gene Ontology. Source data are available online for this figure.

them by analyzing a different secondary variant recurrently identified in LGGNTs (p.T658P, found in combination with K656E in DNETs and pilocytic astrocytoma samples (Jones et al, 2013; Pathak et al, 2017). Similar to K656E/R661P, double mutant K656E/T658P showed reduced protein stability and attenuated activation of MAPK/ERK signaling compared to K656E (Fig. EV3C,D), confirming that modulatory mechanisms mediated by secondary mutations might represent a recurrent phenomenon.

## The R661P variant destabilizes PLCγ binding acquired by N546K mutant

As mentioned above, the proximity interactome revealed differences in the abundance of crucial downstream signaling mediators between WT FGFR1 and oncogenic mutant baits (Fig. 2F). Among those, we decided to further investigate the interaction with PLCγ since it was found strongly increased in both N546K and N546K/R661P captures, pointing to extended binding and activation of downstream signaling (Fig. 5A). PLCγ is encoded by the PLCG1 gene and, upon binding to RTK receptors is activated and mediates the cleavage of hydrolyzed phosphatidylinositol-4, 5-bisphosphate (PIP2) to produce inositol-1,4,5-triphosphate (IP3) and diacylglycerol (DAG), which in turn regulate multiple cellular processes in a tissue-specific manner (Yang et al, 2012). To further understand this interaction at the molecular level, we leveraged the structure prediction algorithm Alphafold3 (AF3) (Abramson et al, 2024) to generate a 3D model of the FGFR1-PLCγ complex (Fig. 5B) and confirmed that the model correctly recapitulates experimentally validated interactions between specific domains (Hajicek et al, 2019) and superposes with the crystalized structure of the FGFR1 active kinase domain (Appendix Fig. S3). In line with our interactome results, the residue Asn546 (contrary to Lys656 and Arg661) is located at the surface of interaction between FGFR1 and PLCγ (Fig. 5B), and the side chain of the mutated residue (lysine) is exposed toward the interface of interaction, suggesting that it might be stabilizing the binding (Fig. 5B). Despite this interaction was also strong in the double mutant N546K/R661P, it appeared attenuated (Fig. 5A), potentially indicating reduced affinity compared to the respective single mutant. The increased ability of the N546K single- and double-mutant protein to bind and phosphorylate PLCγ1 was validated through co-immunoprecipitation experiments (Fig. 5C). Once again, the presence of the secondary mutation R661P was found to weaken this interaction and consequent activation of PLCγ, with lower levels of immunoprecipitated phosphorylated proteins, highlighting another modulation mechanism played by R661P on the activating potential of the N546K mutation.

## R661P severely impairs the proliferative potential of oligodendroglioma cells

It has been proposed that DNET and other types of glioneuronal tumors originate from oligodendrocyte lineages and express molecular signatures of oligodendrocyte precursors (Duan et al, 2024; Luzzi et al, 2019; Matsumura et al, 2013). To study the impact of the mutations investigated so far on a model that could better recapitulate the molecular background of FGFR1-mutated brain tumors we used the human oligodendroglioma cell line (HOG) and we applied CRISPR/Cas9 to generate isogenic FGFR1-mutant HOG cell lines (Fig. 6A). HOG cells have been shown to recapitulate immature oligodendrocyte lineage and have been used in previous reports to study oligodendrocyte function (Buntinx et al, 2003; De Kleijn et al, 2019; Post and Dawson, 1992). They are hemizygous for FGFR1, as they harbor a genomic deletion in chr8p.23 encompassing the FGFR1 locus (Fig. 6A). This aspect was crucial to study FGFR1 mutations without WT gene expression, as it allowed us to force our cellular systems to reveal variant-associated phenotypes. Thus, two independent clones for the same FGFR1 mutations characterized in the interactome profiling (N546K, K656E, R661P, N546K/R661P and K656E/R661P) were isolated and correct editing was confirmed (Appendix Fig. S4). Mutant clones presented no apparent differences in morphology except from R661P-expressing cells, which tend to form more compacted clusters, with cells establishing more surface contacts (Fig. EV4A). Molecularly, this could be explained by a different polarization mediated by FGFR1-R661P at the plasma membrane, engaging interactions with different actin-interacting proteins than the WT protein, as indicated by analysis of the R661P-specific BioID cluster 14 (Fig. EV4B; Dataset EV1). Strikingly, colony formation assays pointed to the R661P germline variant as the mutation with the most severe effect, almost completely abolishing the clonogenic potential of HOG cells (Fig. 6B). In these assays, both double mutants displayed an intermediate phenotype between the one caused by the germline variant and their respective oncogenic single mutants (Fig. 6B).

To study whether the R661P proliferative defect was associated with a loss of FGFR1 function, we also generated a knock-out (KO) clone (Fig. EV4C). Although FGFR1-KO cells showed impaired self-renewal ability compared to the parental cell line, this effect was less pronounced than for R661P-mutant cells (Fig. 6C). This observation suggests that the phenotypic consequences of the R661P variant are probably linked to a change, rather than a loss, of receptor function, which leads to more dramatic effects in HOG

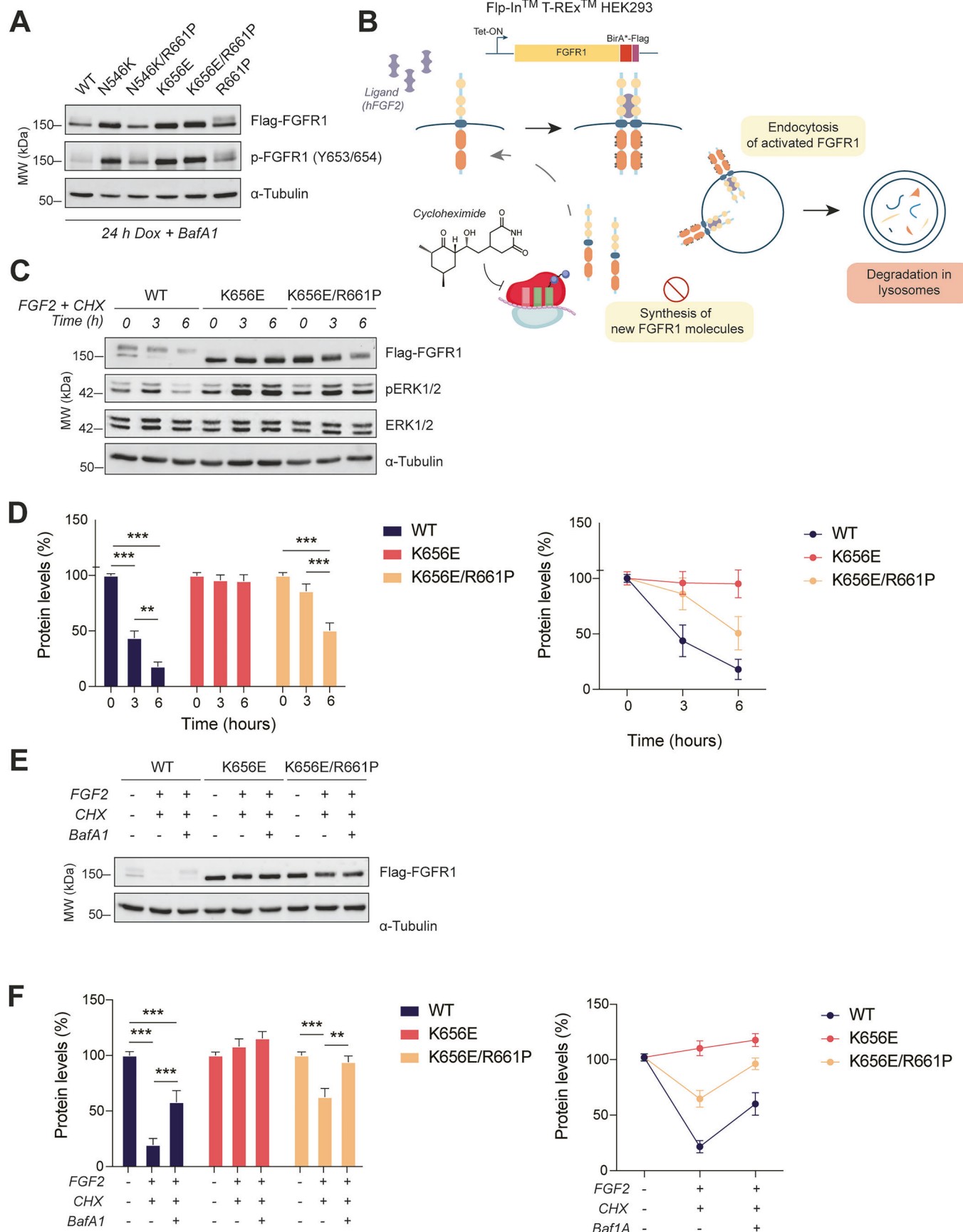

**Figure 4.  R661P reduces oncogenic mutant protein stability by rescuing lysosomal degradation.**

(A) Western blot revealing Flag-FGFR1 and p-FGFR1 levels in cells after 24 h of Tet induction with doxycycline and parallel treatment with the lysosome inhibitor Bafilomycin A1 (BafA1). (B) Receptor stability experimental design: HEK293 cells expressing WT and mutant FGFR1 are stimulated with human FGF2 (hFGF2, 25 ng/mL) to induce receptor internalization and degradation and, in parallel, with cycloheximide (100 µg/mL) to block protein synthesis. (C) Western blot analysis of a representative time-course experiment comparing WT FGFR1, K656E single and double mutants showing total Flag-FGFR1 as well as total and phospho-ERK protein levels at 0, 3 and 6 h post-treatment with FGF2 and cycloheximide (CHX). (D) Relative amounts of WT, K656E single and double mutant FGFR1 protein obtained by quantifying western blots from $n = 3$ independent experiments. Values have been normalized (FC) against each relative 0 h reference values. Data is represented by mean ± SEM and significant variations in protein amounts against each specific reference values have been indicated in the bar plots (ANOVA test); $P$ values from left to right: ***$P = 4.9e\text{-}12$, ***$P < 1e\text{-}12$, **$P = 0.0026$, ***$P = 7.4e\text{-}09$, ***$P = 4.0e\text{-}05$. (E) Western blot analysis of WT FGFR1, K656E and K656E/R661P mutants indicating total Flag-FGFR1 protein levels upon FGF2 induction and treatment with only cycloheximide (100 µg/mL), or with cycloheximide combined with Bafilomycin A1 (200 nM) for 6 h. (F) Relative amounts of WT, K656E single and double mutant FGFR1 protein obtained by quantifying western blots from $n = 4$ independent experiments. Values have been normalized (FC) against each relative 0 h reference values. Data is represented by mean ± SEM and significant variations in protein amounts against each specific reference values have been indicated in the bar plots (ANOVA test); $P$ values from left to right: ***$P = 2.2e\text{-}10$, ***$P = 0.00011$, ***$P = 0.00053$, ***$P = 0.0009$, **$P = 0.00993$. Source data are available online for this figure.

cells compared to total loss of *FGFR1* expression. Finally, we tested a FGFR1 WT-clone isolated during CRISPR experiments, which exhibited no significant differences with the parental cell line (Fig. 6C). This result provides additional evidence that the observed phenotype was mutation-specific, and not a stochastic fluctuation of proliferation rates in HOG cell population. Defective self-renewal ability of the two R661P clones was confirmed in 3D-Matrigel colony formation assays, with a reduction in the size of colony area of 70–80% (Fig. 6D). To exclude cell line-specific effects, we sought to validate these results in another cell model with oligodendroglial features. We generated FGFR1-R661P clones using MO3.13 cells (Appendix Fig. S4), an immortalized cell line which have been extensively used in literature as a model of immature oligodendrocyte, to study oligodendrocyte function (Del Grosso et al, 2016; Hoshino et al, 2020; McLaurin et al, 1995). FGFR1-R661P MO3.13 clones showed reduced self-renewal capacity compared to WT cells (Fig. EV4D), supporting the results obtained with HOG cells. In summary, these results unveiled severe defects in the proliferative potential of oligodendroglioma cells caused by the R661P mutation.

### R661P-mutant oligodendroglioma cells present changes in developmental gene expression, recovered in double mutant cells

To investigate the molecular events driving the phenotypic changes observed above in HOG cells, we performed differential gene expression analysis through RNAseq using the two R661P mutant HOG clones and the parental cell line, identifying $n = 338$ of differentially expressed (DE) genes shared by the two clones, representing the 22% and 26% of total DE genes for clone #1 and clone #2, respectively (Fig. 7A; Dataset EV2). Gene ontology analysis of DE genes in FGFR1-R661P HOG cells revealed enrichment for development-related categories, some of them specific for brain tissue formation and differentiation (Neural Crest Cell development, Axon guidance, Axonogenesis), alongside with cellular processes (Epithelial cell proliferation, Negative regulation of cell adhesion and Cell-matrix adhesion) which may underlie the R661P cell phenotype discussed in the previous section (Fig. 7B). We selected key genes belonging to these categories that were found dysregulated in R661P clones and we validated them through RT-qPCR (Fig. 7C). These genes encode for described master transcriptional regulators (*SOX9, GLI2, HIF1A*), signaling factors (*WNT5A, NRP1, SEMA3E*), and surface glycoproteins (*THBS1, FN1*) with central roles during development, neuro-glial cell fate

decisions and oligodendrocyte differentiation (Allan et al, 2021; Blake et al, 2008; Endo and Minami, 2018; Finzsch et al, 2008; Hassel et al, 2023; Oh and Gu, 2013; Qi et al, 2003; Sherafat et al, 2021).

To assess whether the mentioned processes are also altered in other FGFR1 mutants, or they are R661P-specific, we also performed RNAseq using N546K, K656E and double mutants N546K/R661P and K656E/R661P clones. By extending the number of categories of interest obtained in Fig. 7B to more general development-related categories, we plotted differential gene expression values obtained by comparing each mutated clone with the parental cell line, focusing on unique genes belonging to these groups (Fig. 7D). Despite the presence of the R661P mutation, double mutant N546K/R661P and K656E/R661P HOG cells showed a regression toward parental expression levels, with a profile close to the one traced by N546K and K656E clones (Fig. 7D). These observations indicate that the presence of the hotspot *in cis* is able to compensate for potential defects caused by the constitutional allele and point to a novel model of susceptibility for the *FGFR1* allele R661P.

## Discussion

Multiple mutations *in cis* in tumor driver genes are rare events, mostly associated with recurrence and acquired therapy-resistant phenotypes (Barber et al, 2013; Castaneda-Gonzalez et al, 2022; Kondrashova et al, 2017; Stewart et al, 2015). By interrogating GENIE project data, the largest tumor cohort publicly available so far, we confirmed the recurrence of secondary variants along with K656E and N546K hotspots in brain tumors, more frequently found with the first one. Next, we employed complementary models and strategies, providing multiple evidence on both molecular mechanisms and phenotypes triggered by double mutant FGFR1 receptors. FGFR1 and, in general, RTK signaling is spatially and temporally controlled by fine-regulated intracellular vesicular trafficking (Goh and Sorkin, 2013; Hinsch et al, 2023; Miaczynska, 2013). The data presented in this study highlights an oncogenic mechanism for the two hotspot mutations that relies on high tyrosine kinase activity and altered protein transport and evasion of degradation processes, resulting in the accumulation of active (phosphorylated) FGFR1 receptor in cells. Among all FGFR1 oncogenic mutations investigated in this work, N546K represents the one with the higher autophosphorylation profile, in line with a described biochemical model that proposed a faster and ligand-

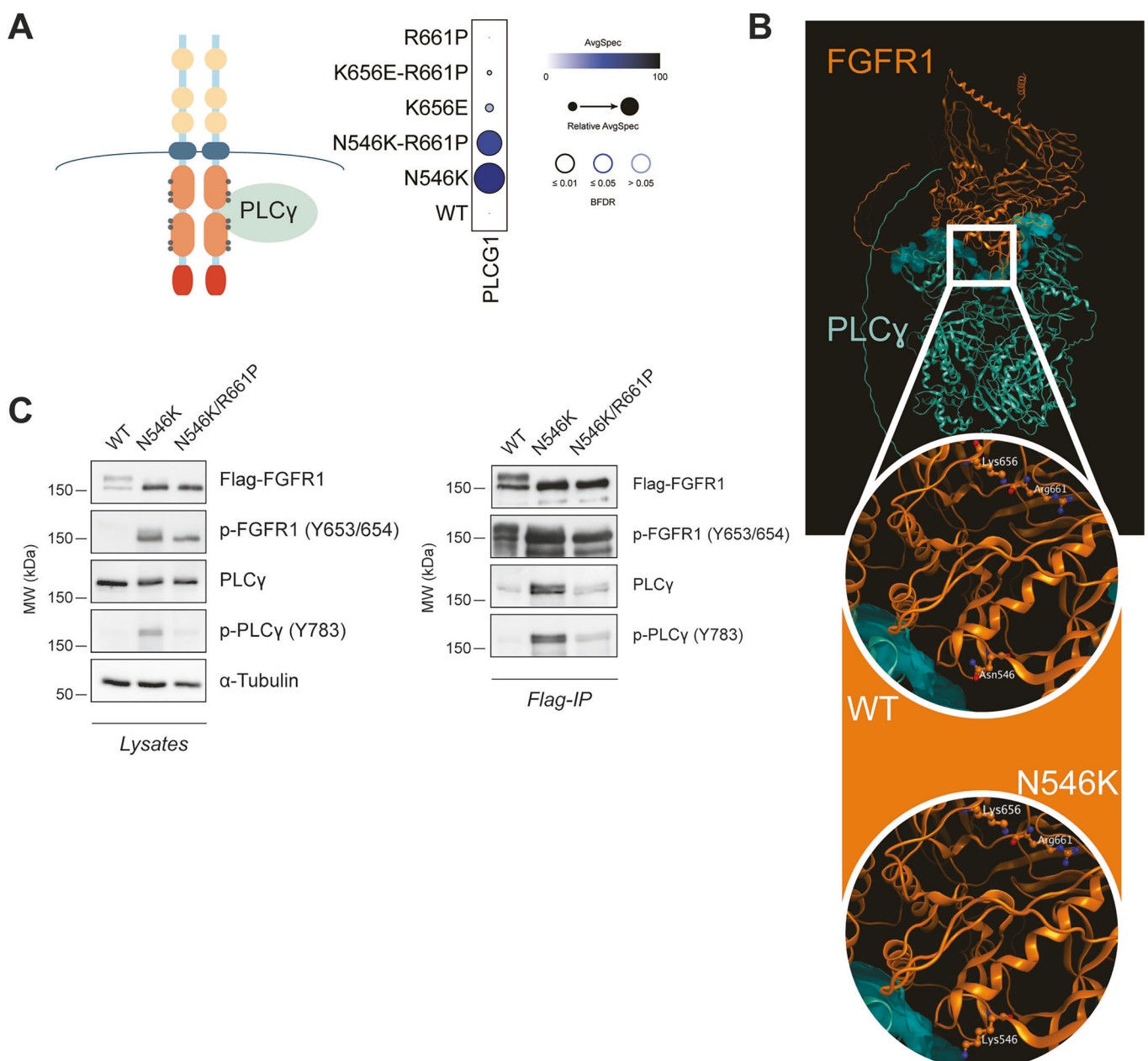

**Figure 5. N546K/R661P double mutant interacts with PLCγ with reduced affinity compared to N546K.**

(A) Schematic representation of activated FGFR1 bound to PLCγ (left) and dot plot with relative abundances (AvgSpec) of PLCγ protein for WT and mutant FGFR1 obtained through the BioID-MS screening. (B) Predicted 3D structure of FGFR1-PLCγ complex obtained using AF3. The square highlights a zoomed area with the 546 residue in WT (Asn) and mutant (Lys) FGFR1 protein, locating in the interface of FGFR1-PLCγ interaction. The residues Lys656 and Arg661 are also highlighted, to show how residue 546 is the only one close to the interface of interaction and supporting a role for the N546K mutation in enhancing the binding between the two proteins. (C) Western blot of total lysates and the co-immunoprecipitation (IP) of total and phosphorylated FGFR1 and PLCγ proteins, using the T-REx HEK293 stable cell lines. AvgSpec average of spectral counts. Source data are available online for this figure.

independent autophosphorylation (Lew et al, 2009), which could mediate the activation of downstream signaling pathway and, in parallel, the escape of degradation mechanisms.

Little experimental evidence is available in literature for K656E activating molecular features (Fisher et al, 2021; Hart et al, 2000). However, the analogous variant in FGFR3 (K650E) in germline carriers causes skeletal dysplasia syndromes (Foldynova-

Trantirkova et al, 2012; Wilcox et al, 1998), and has been reported to enhance autophosphorylation and induce proliferative phenotypes in cells (Bellus et al, 2000; Monsonego-Ornan et al, 2002; Naski et al, 1996). Notably, ectopic expression of FGFR3 K650E resulted in a similar loss of the upper band in western blot experiments, indicating analogous changes in the combination of post-translational modifications induced by the amino acid change

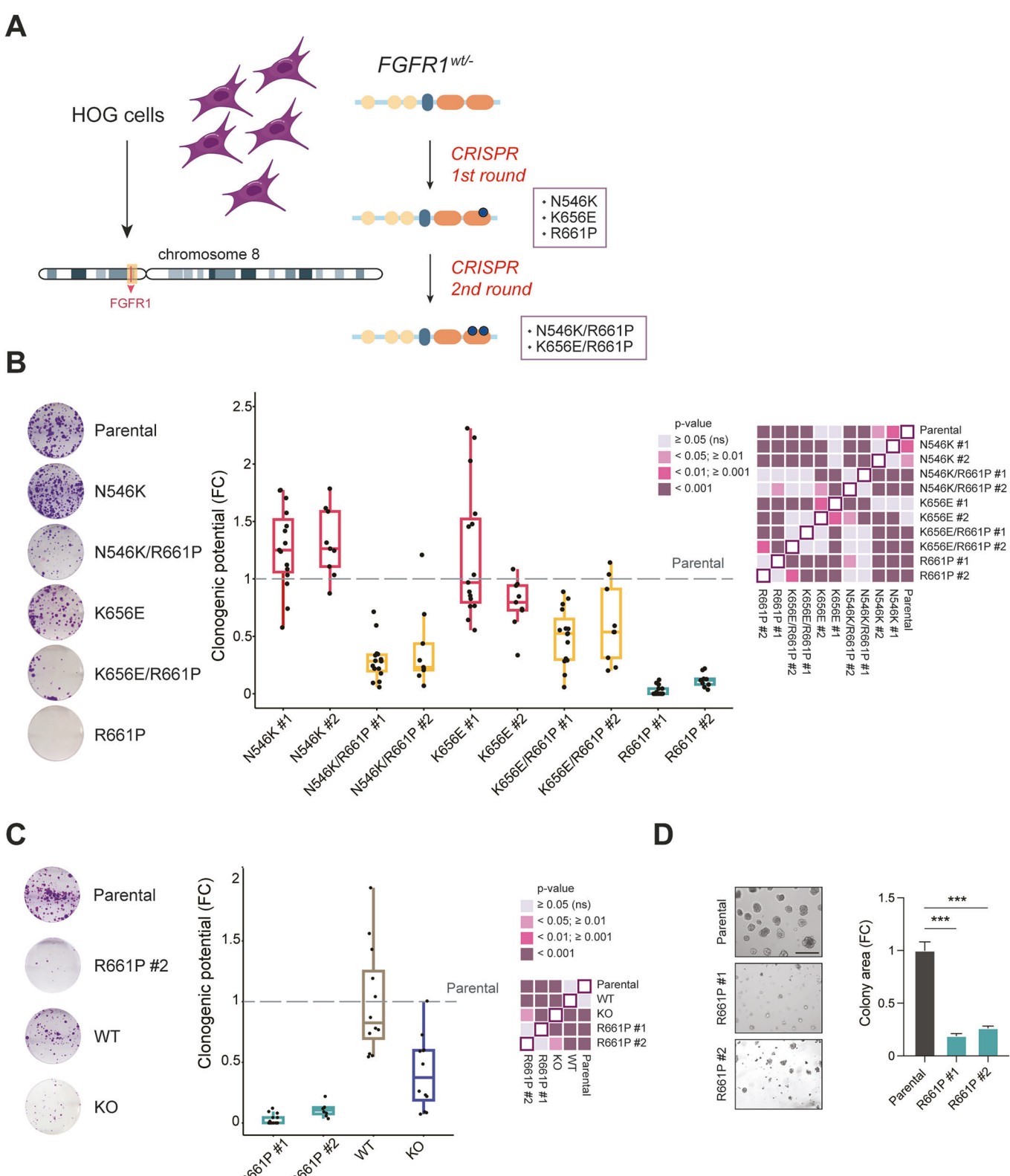

(Bellus et al, 2000; Monsonego-Ornan et al, 2002). In fact, FGFR1 Lys656 was included in a screening of multiple FGFR1 lysine residues mutants, which resulted in escaped lysosomal degradation (Haugsten et al, 2008), reinforcing our premise that oncogenic mutants affecting this conserved residue act through dysregulation of degradation pathways in FGF receptors.

For pediatric and epileptogenic low-grade glioneuronal tumors such as DNETs, tumor resection is the treatment of choice.

◄ **Figure 6. Defects in self-renewal and changes in phenotype imposed by R661P mutation.**

(A) Schematic representation of the CRISPR design to generate FGFR1 single- and double-mutant clones using the HOG cell line. Genomic deletion in *FGFR1* locus in chromosome 8 (chr.8p11.23) identified in these cells has been represented. (B) Colony forming assays comparing CRISPR-edited, FGFR1-mutant HOG clones. Both clones for each mutation (N546K, K656E, N546K/R661P, K656E/R661P and R661P) have been assayed. Microscope captures of representative wells of clones #1 for each mutant have been included on the left and quantification of $n \geq 3$, experiments have been plotted, normalized against the parental cell line (dashed line). Boxes represent the interquartile range (IQR) from the 25th percentile (Q1) to the 75th percentile (Q3). The horizontal line represents the median and whiskers extend to the smallest and largest values, outliers excluded. Replicates are represented by black dots. Results of statistical analysis of multiple comparisons (ANOVA test) have been plotted in the heatmap on the right. (C) Colony forming assays comparing the two R661P clones with an WT clone and a KO clone. Representative wells of the second R661P clone, WT and KO clones are indicated. Quantification of $n \geq 3$, experiments have been plotted, normalized against the parental cell line (dashed line). Boxes represent the interquartile range (IQR) from the 25th percentile (Q1) to the 75th percentile (Q3). The horizontal line represents the median and whiskers extend to the smallest and largest values, outliers excluded. Replicates are represented by black dots. Results of statistical analysis of multiple comparisons (ANOVA test) have been plotted in the heatmap on the left. (D) Matrigel 3D colony forming assays comparing the two R661P clones with the parental cell line. Captures of a representative field displaying colonies formed at experimental endpoint (day 7) are reported on the left. Scale bar: 200 μm. On the right: bar plot of quantified colony areas, normalized against control condition (parental cell line). Mean ± SEM values ($n \geq 3$ independent experiments) and significant differences have been indicated (Kruskal–Wallis test); *P* values from left to right: ***$P = 1.9e$-05, ***$P = 4.6e$-06. Source data are available online for this figure.

However, surgery is not always viable due to the difficulties in excising tumors and efficient resection is often limited by unclear margins (Devaux et al, 2017). Moreover, despite their benign character, tumor relapse in DNETs and other LGGNTs is not rare, and highly associated to incomplete resection and seizure outbreaks (Chassoux et al, 2012; Nolan et al, 2004), highlighting a strong need for novel therapeutic approaches. FGFR inhibitors have been tested in clinical trials to treat aggressive cancers harboring alterations in FGFR genes, including cases with FGFR1 alterations, with only partial responses and in some cases severe adverse events (Pant et al, 2023; Subbiah et al, 2022). Partial explanation might come from their non-specific nature, as all FGFR inhibitors previously or currently tested in clinical trials also target other RTKs or other members of the FGFR family. Moreover, in vitro screening of a panel of FGFR inhibitors revealed mutation-specific responses (Nakamura et al, 2021), and recent findings comparing FGFR1 structural variants- and N546K-expressing cell models showed differences in sensitivity to FGFR inhibitors (Apfelbaum et al, 2025), highlighting the importance to identify mutation-specific vulnerabilities in future drug discovery strategies (Nakamura et al, 2021). In the last few years, the spectrum of FGFR inhibitors has increased, including new molecules that specifically target FGFR1, which have not been fully tested in pre-clinical models yet (Fan et al, 2024). On the other hand, different drug design approaches, targeting protein stability, such as proteolysis-targeting chimeras (PROTACs), an expanding field in oncotherapy (Vikal et al, 2025) with already available first candidates targeting specifically FGFR1 (Wang et al, 2024), might be taken into consideration for testing in N546K and K656E mutated tumors.

Aside from the two hotspots mutations discussed so far, our work provides extensive data on the susceptibility allele R661P (Rivera et al, 2016). FGFR1 R661P-mutant protein clusters close to the WT protein in interactome profiles, suggesting less impact on FGFR1 protein function, compared to the oncogenic mutants. This observation is coherent with the absence of pathological effects other than multinodular DNETs in R661P carriers, who had been clinically followed until adulthood. Most likely in tissues other than brain this variant alone does not lead to major changes in the receptor function, which distinguishes it functionally from other germline *FGFR1* variants that cause developmental conditions. Nevertheless, our results indicate that the R661P variant does have dramatic effects in terms of self-renewal ability and rewiring of gene expression at the target tissue. A similar strong genotype-

phenotype specificity at the tissue type-level is observed for germline variants causing osteoglophonic dysplasia and giant cell tumors of the jaw (Gomes et al, 2018; White et al, 2005). Consistent with the tissue-specificity of the biological effects associated with the R661P variant, differential gene expression analysis revealed enrichment for neural development categories with marked down-regulation of key transcription factors involved in glia cell development and oligodendrocyte differentiation, as *SOX9* (Finzsch et al, 2008; Hassel et al, 2023; Pozniak et al, 2010). Notably, a similar association with neuronal transcriptional categories has been recently reported for both *FGFR1*-mutated pediatric low-grade gliomas and neural precursors cell lines (Apfelbaum et al, 2025).

*FGFR1* is expressed in glial and neural precursors and plays essential functions during brain tissue development and differentiation of neural cell lineages, including oligodendrocyte precursor cells (Furusho et al, 2011; Grabiec et al, 2016; Yoon et al, 2004). In this context, both the oligodendrocyte lineage background and the absence of WT allele expression in R661P HOG cells would contribute to reveal the phenotypic defects caused by the mutant receptor. Importantly, we showed here that double mutant K656E/R661P and N546K/R661P rescue both stemness/proliferative capacity and expression levels of development-related genes, increasing the repertoire of mutation-specific features that characterize FGFR1 mutants. The R661P mutation in combination with the K656E is also found in one case (likely sporadic, based on their variant allele frequency) from the GENIE cohort, arguing that the combination of these two events perpetuates tumorigenesis. These results may be unveiling a novel model of oncogene-associated susceptibility, based on the need for "correction" of proper signaling and transcriptional programs, which ultimately predisposes toward selection of oncogenic events conferring driving capacity to the mutated clone. Sporadic cases harboring other missense mutations (as the double mutant K656E/T658P) might share common mechanisms with the ones uncovered in the present work, thus enlightening why DNETs and other therapy naive gliomas with a quiet genome acquire additional hits involving the *FGFR1* gene.

In hereditary tumor syndromes, a continuum model for tumor suppressors evolves the classical two-hit model to a fine-tuning sensitivity of the target cell to either survive, senesce or become tumorigenic depending on the effects of mutational pattern at each tissue level (Albuquerque et al, 2002; Berger et al, 2011; Latchford

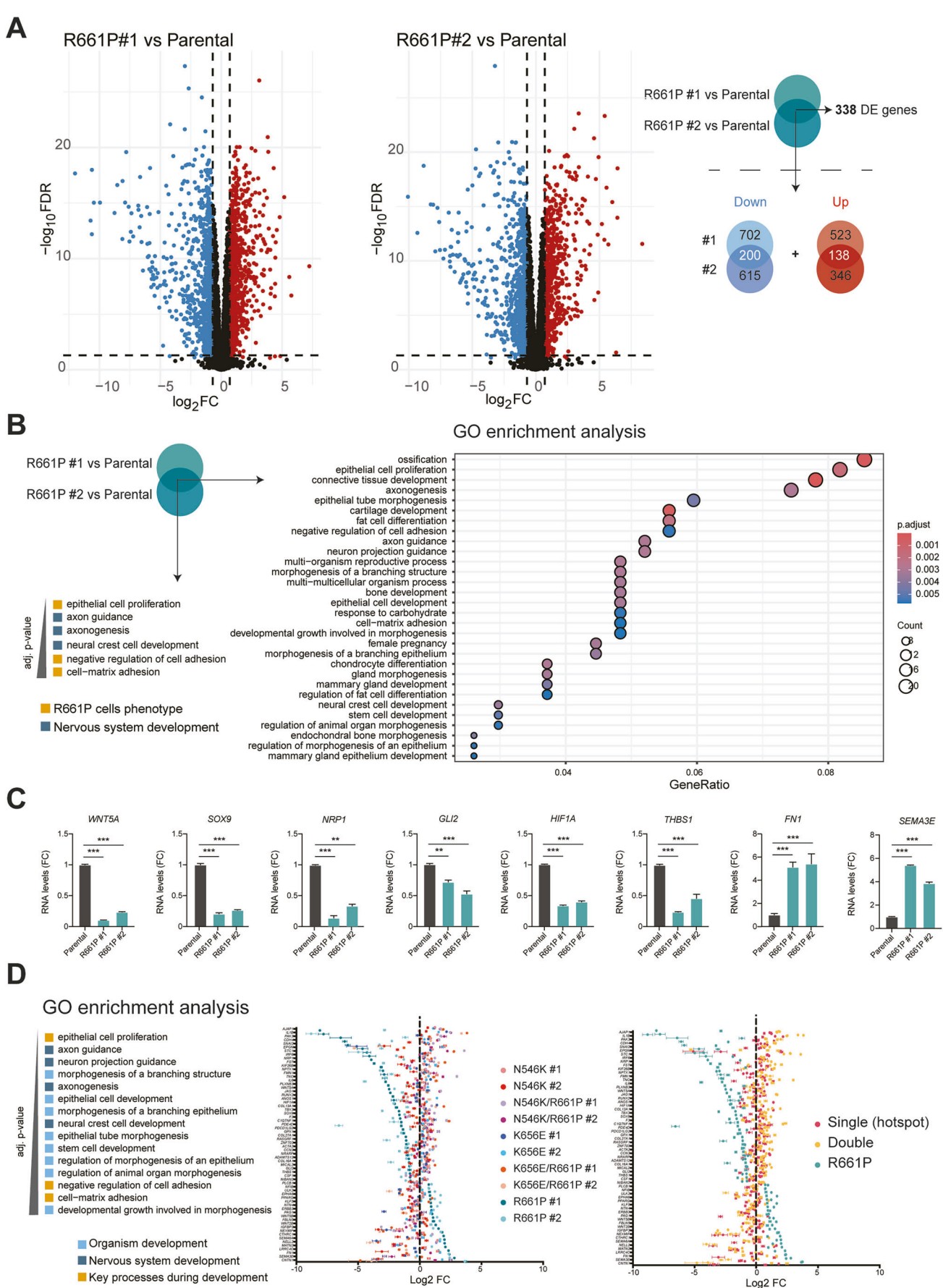

◄ **Figure 7.  R661P mutant HOG cells show dysregulation of development and neurodevelopment-associated genes, rescued in double mutants.**

(A) Volcano plots representing downregulated (red dots) and upregulated (blue dots) genes differentially expressed genes comparing each R661P clone ($n = 3$ replicates) with the Parental cell line ($n = 3$ replicates). Log fold-change is plotted on the $x$ axis and the adjusted $P$ value (FDR; –log10 scale) on the $y$ axis. Dashed horizontal and vertical lines indicate threshold of significance (FDR ≤ 0.05) and absolute fold change (>0.7). Numbers of specific and common DE genes are specified in Venn diagrams (right). (B) Dot plot of GO enrichment analysis of R661P vs Parental differentially expressed (DE) genes, using common DE genes obtained from both comparisons shown in (A)) as input list. Significantly enriched (adj. $P$ value ≤ 0.05, Benjamini–Hochberg method) have been represented. Size and color of dots represent number of matched genes and adj. $P$ value, respectively. (C) RT-qPCR validation of key genes, comparing RNA levels in R66P1 clones vs Parental HOG cells. Mean ± SEM values from $n = 2$ independent experiments have been plotted and significant differences have been indicated (ANOVA test); $P$ values from left to right: *WNT5A* \*\*\*$P < 1e-12$, \*\*\*$P < 1e-12$, *SOX9* \*\*\*$P = 2.1e-12$, \*\*\*$P = 3.1e-11$, *NRP1* \*\*\*$P = 1.8e-06$, \*\*$P = 0.00234$, *GLI2* \*\*$P = 0.00472$, \*\*\*$P = 6.0e-06$, *HIF1A* \*\*\*$P < 1e-12$, \*\*\*$P < 1e-12$, *THBS1* \*\*\*$P = 3.0e-08$, \*\*\*$P = 2.6e-05$, *FN1* \*\*\*$P = 3.2e-07$, \*\*\*$P = 3.1e-07$, *SEMA3E* \*\*\*$P < 1e-12$, \*\*\*$P < 1e-12$. (D) Selected GO categories for further analysis are illustrated. Dot plots representing Log$_2$FC values of RNAseq DE analysis obtained comparing each FGFR1 mutant clone with the Parental cell line. Error bars represent SD among triplicates. Selected genes ($n = 64$) that have been plotted are unique genes identified in the GO categories listed on the left (selected from GO in (B)). Left plot: each independent clone has been represented with a different color. Right plot: Single mutant clones (hotspots N546K #1 and #, K656E #1 and #2) have been represented in red; double mutant clones (N546K/R661P #1 and #2, K656E/R661P #1 and #2) have been represented in light orange and R661P #1 and #2 clones have been represented in blue. DE differentially expressed. Source data are available online for this figure.

et al, 2007). Similarly, a most-optimal level of FGFR1 signaling during early onset tumorigenesis might prevail, with additional layers of complexity due to the key roles of the FGF-FGFR axis plays during development (White et al, 2005).

Here we present evidence for modulating effects exerted by *FGFR1* secondary mutations on recurrent hotspot oncogenic mutations, which combined could provide a tolerated level of FGFR1 signaling required during brain development and, on the other hand, trigger oncogenic processes that cause early-onset tumor formation. Future efforts are necessary to confirm this modulatory ability for other missense variants co-identified with the two hotspot mutations. The propensity of certain tumor types to acquire second alterations without any external (treatment) pressure refers to the plasticity of the tumor cells to diverge and adapt through tumor development. Understanding this process might forecast tumor response to external pressures imposed by targeted therapies. The present study provides a testimony of novel oncogenic mechanisms and modulatory effects derived from multiple *FGFR1* alterations that will be informative when defining mutation-specific treatment options and might help to identify novel therapeutic approaches for these patients.

# Methods

### Reagents and tools table

| Reagent/resource | Reference or source | Identifier or catalog number |
|---|---|---|
| **Experimental models** | | |
| HEK293 T-REx Flp-In (Human) | Thermo Fisher | #R78007 |
| HOG (Human) | Dr. Lopez Guerrero Lab | |
| MO3.13 (Human) | Dr. Querol Gutierrez Lab | |
| **Recombinant DNA** | | |
| pDONR-223 vector | Addgene | |
| pcDNA5-FRT backbone vector | | |
| Flippase-expressing plasmid (pOG44) | | |
| PX458 vector | Addgene | #48138 |

| Reagent/resource | Reference or source | Identifier or catalog number |
|---|---|---|
| **Antibodies** | | |
| Anti-FLAG | Merck | #F3165 |
| Anti-FGFR1 | Abcam | #ab76464 |
| Anti-phospho-FGFR1 Tyr653/654 | Cell Signaling Technology | #52928 |
| Anti-PLCγ1 | Cell Signaling Technology | #5690 |
| Anti-ERK1/2 | Cell Signaling Technology | #9102S |
| Anti-phospho-ERK1/2 (Thr202/Tyr204) | Cell Signaling Technology | #9101S |
| Anti-phospho-PLCγ1 Tyr783 | Cell Signaling Technology | #2821 |
| Anti-tubulin | Merck | #F6199 |
| Streptavidin-HRP | Thermofisher | #21126 |
| Anti-rabbit-HRP | Merck | #A0545 |
| Anti-mouse-HRP | Invitrogen | #32430 |
| Anti-mouse Alexa Fluor 488 | Invitrogen | #A21202 |
| Anti-rabbit Alexa Fluor 555 | Invitrogen | #A21430 |
| **Oligonucleotides and other sequence-based reagents** | | |
| PCR primers | This study | Appendix Table S2 |
| qPCR primers | This study | Appendix Table S5 |
| gRNA sequences | This study | Appendix Table S3 |
| Alt-R™ HDR Donor Oligos | This study | Appendix Table S4 |
| **Chemicals, enzymes and other reagents** | | |
| Doxycycline | Merck | #D9891 |
| FGF2 | RD Systems | #233-FB |
| Cycloheximide | Merck | C7698 |
| Bafilomycin A1 | Merck | #B1793 |
| Hygromycin B | Merck | #10843555001 |
| Blasticidin | Merck | #SBR00022 |
| Tetracycline | Merck | #T7660 |
| Biotin | Merck | #B4501 |
| Protein G Mag Sepharose beads | Cytiva | #28944008 |

                                                                 

| Reagent/resource | Reference or source | Identifier or catalog number |
|---|---|---|
| Prolong Gold Antifade Reagent with DAPI | Invitrogen | #P36935 |
| Matrigel® | Corning | #354234 |
| T7 endonuclease I | New England Biolabs | #M0302S |
| AZD4547 (Fexagratinib) | Astra Zeneca | 1035270-39-3 |
| **Software** | | |
| R software v4.3.3 | | https://www.r-project.org/ |
| ImageJ software | | https://imagej.net/ij/ |
| SAINTexpress v3.6.1 | Teo et al, 2014 | |
| AlphaFold 3 | DeepMind | |
| STAR v2.7.0a | | |
| limma package version 1.8.2 | Bioconductor | |
| **Other** | | |
| TransBlot Turbo RTA Transfer Kit | Bio-RAD | #1704158 |
| Clarity Western ECL Substrate Kit | Bio-RAD | #1705060 |
| BCA Protein Assay Kit | ThermoScientific | #23227 |
| RNeasy Kit | Qiagen | #74104 |
| SuperScript Reverse Transcriptase II Kit | Invitrogen | #18064014 |
| PowerUp SYBR Green Master Mix | Applied Biosystems | #A25776 |
| RNA 6000 Nano Assay | Agilent | #50671511 |
| TruSeq® Stranded mRNA LT Sample Prep Kit | Illumina | #20020594 |

### Review of public tumor data

Public tumor data was interrogated using the GENIE/cBioportal database (de Bruijn et al, 2023) (https://genie.cbioportal.org/), selecting the GENIE Cohort v16.1 (last accessed in October 2024) and query for *FGFR1* gene. Mutation data was filtered for samples harboring hotspot mutation in one of the codons N546 and K656 and tables with clinical and genomic data was obtained. Data was manually reviewed and second mutations in *FGFR1* were annotated. Mutated samples were divided into brain tumor categories vs others. Mutated samples corresponding to brain tumor types included: Low-Grade Glioma, NOS; Dysembryoplastic neuroepithelial tumor; Pilocytic Astrocytoma; Rosette-forming Glioneuronal Tumor of the Fourth Ventricle; Pilomyxoid Astrocytoma; Miscellaneous Neuroepithelial Tumor; Glioma, NOS; Ganglioglioma; Miscellaneous Brain Tumor, Diffuse Glioma; Glioblastoma Multiforme; High-Grade Glioma, NOS; Anaplastic Oligodendroglioma; Anaplastic Astrocytoma; Diffuse Intrinsic Pontine Glioma; Glioblastoma; Oligodendroglioma; Primary Brain Tumor; Astrocytoma; Encapsulated Glioma). Available data from brain tumor samples was reviewed and tumor grading "low" (for grade 1-2) or "high" (for grade 3–4) was assigned when possible, according to the annotated clinical diagnosis and molecular data

following the WHO-IARC Classification of tumors, Central Nervous System, 5th Edition (Louis et al, 2021). Tumor cases with tumor grade which was not possible to categorized into low- or high-grade were considered as unknown.

### Cell cultures

The Flp-In T-REx HEK293 (Thermo Fisher®) cell line was purchased from Thermo Fisher Scientific, while the Human Oligodendroglioma (HOG) cell line and the MO3.13 hybrid cell line were generous gifts from Dr. Lopez Guerrero and Dr. Querol Gutierrez, respectively. Cell lines have been typified and regularly tested for mycoplasma contamination every two months. Cultures were maintained in Dulbecco's Modified Eagle Medium (DMEM) with 4.5 g/L glucose (Gibco, Thermo Fisher), supplemented with 10% fetal bovine serum (FBS) (Gibco, Thermo Fisher) and 1% penicillin–streptomycin (P/S) (10,000 U/mL) (Gibco, Thermo Fisher), at 37 °C in a 5% $CO_2$ atmosphere. Passages were performed once cells reached 80-90% confluency, by trypsinization and subsequent dilution into fresh culture plates.

To induce expression of FGFR1 receptors, Flp-In T-REx HEK293 were treated with 1 µg/mL of Doxycycline (Merck) for 24 h in every experiment described, except for time course experiment where cells were treated for 8 h (Fig. 3D–F). For protein stability and degradation experiments, cells were starved overnight with FBS-free DMEM. Next, cells were induced with 25 ng/mL of recombinant human FGF2 (RD Systems) and treated with 100 µg/mL of Cycloheximide (Merck) and 200 nM of Bafilomycin A1 (Merck). For tyrosine kinase inhibition experiments, cells were treated four hours after Tet-induction (Doxycycline 1 µg/mL) either with DMSO or AZD4547 (Fexagratinib) compound (10 nM). Lysates were collected at 24 h post-Tet induction.

### Generation of stable cell lines and BioID-MS

FGFR1 WT cDNA sequence was recombined into the pDONR-223 vector and then shuttled into the pcDNA5-FRT backbone vector expressing an abortive mutant of BirA (BirA*) tagged with Flag using the gateway recombination cloning system (Hartley et al, 2000). Stable and inducible Flp-In T-REx HEK293 cell lines expressing FGFR1- BirA*-Flag or Empty-Flag-BirA* or Empty-3xFlag under the tetracycline operator (Tet), were generated according to the manufacturer protocol. Briefly, cells were transfected with the FGFR1-BirA*-FLAG constructs along with a Flippase-expressing plasmid (pOG44), to induce recombination and integration in the cells' genome. Cells were then selected using 200 µg/mL Hygromycin (Merck) and 5 µg/mL Blasticidin for three weeks. For BioID-MS experiment, cells were seeded in 15 cm plates to reach 70–80% confluency. The next day, cells were treated with tetracycline (1 µg/µL) and biotin (50 µM, Merck). Cells were then processed for MS analysis as previously described (Hesketh et al, 2020). Validation of biotinylation of proximal interactors was carried out through western blot using the anti-streptavidin-HRP antibody (see "Western blotting" section).

### BioID data analysis

Reproducibility of samples among biological replicates was assessed in R (r-project.org) using the stats package to perform Spearman correlations on spectral counts. To calculate interaction statistics,

we used SAINTexpress (Teo et al, 2014) version 3.6.1 on proteins with an iProphet protein probability $\geq 0.99$ and unique peptides $\geq 2$. Proteomics datasets were compared separately against negative controls ($n = 22$). SAINT analyses were conducted by compressing $n = 11$ controls (Empty- BirA*-Flag and Empty-3xFlag) to the 11 highest spectral counts among the 22 samples, while baits were not compressed. To infer FGFR1's high confidence interacting preys, we applied a combination of filters. Preys had to display a SAINT average probability (AvgP) $\geq 0.95$, which is more stringent than a BFDR $\leq 0.01$ threshold, and an average spectral count (AvgSpec) $\geq 5$. Unfiltered contaminants, such as Keratin, BirA*, carboxylases and beta galactosidase were manually removed. We assessed the interaction specificity to each bait by calculating their WD-score with the SMAD R package (Sowa et al, 2009). SAINT scoring was performed on a larger dataset that included localization controls and other mutants not discussed explicitly in the text; all raw data and information can be found on ProteomeXchange through partner MassIVE (MSV000096690). Heatmaps were created with the R complexheatmap package (Gu et al, 2016). To do so, we first created a matrix containing the AvgSpec of each bait-prey interaction. Unidentified interactions were input with an AvgSpec of 0, and a pseudocount of 1 was added to the matrix. The data was log2-transformed, and we calculated Canberra distances between filtered preys and Pearson correlations between baits. Using the Ward. D method, we then extracted 17 clusters of preys and two clusters of baits. Optimal numbers of extracted clusters were estimated on Canberra distances by the silhouette method from the factoextra R package and its fviz_nbclust function. Interaction recalls of FGFR1 were extracted from the human BioGRID interaction database version 4.4.236 (Last access August 2024). Gene ontology (GO) was performed using the clusterProfiler (version 3.0.4) R package, correcting for False Discovery Rate (FDR) through the Benjamini–Hochberg method. GO plots were generated through the ggplot2 (3.5.1) R package.

## Western blot

Cells were lysed with 100 μL of SDS lysis buffer (25 mM Tris pH 8, 1 mM EDTA, 1% SDS, 10 mM sodium pyrophosphate in dH$_2$O). The lysates were incubated at 95 °C for 20 min and centrifuged at maximum speed for 10 min. Supernatants were collected and protein concentration was determined using a BCA Protein Assay Kit (ThermoScientific).

For western blotting, 30 μg of protein was aliquoted and mixed with loading buffer (150 mM Tris pH 6.8, 6% SDS, 15% glycerol and 6% β-mercaptoethanol in dH$_2$O). Samples were run on an SDS polyacrylamide gel. Subsequently, the proteins were transferred to a nitrocellulose membrane using the TransBlot Turbo RTA Transfer Kit (Bio-Rad) according to the manufacturer's protocol. The transferred membranes were blocked in either 5% milk or 5% BSA (depending on the antibody requirements) and incubated with primary antibody overnight at 4 °C. The primary antibodies used were anti-FLAG (Merck #F3165), anti-FGFR1 (Abcam #ab76464), anti-phospho-FGFR1 Tyr653/654 (CST #52928), anti-PLCγ1 (CST #5690), anti-phospho-PLCγ1 Tyr783 (CST #2821), anti-ERK1/2 (CST #9102S), anti-phospho-ERK1/2 Thr202/Tyr204 (CST #9101S), anti-tubulin (Merck #F6199) and anti-streptavidin-HRP (Thermofisher #21126). After primary antibody incubation, membranes were washed three times for 5 min with TBST buffer

(100 mM Tris, 150 mM NaCl and 0.1% Tween-20 in dH$_2$O) and incubated with HRP-conjugated secondary antibodies at room temperature for 1 h. The secondary antibodies used were anti-rabbit-HRP (Merck #A0545) and anti-mouse-HRP (Invitrogen #32430). Then, membranes were washed again three times for 5 min with TBST buffer and protein bands were detected using Clarity Western ECL Substrate Kit (BioRad). Images were taken using an Amersham Imager 600 and densitometry analysis was performed using ImageJ software. Three densitometry measurements for each independent experiment have been carried out and outputs have been corrected using loading control (α-tubulin) values. CST = Cell Signaling Technology.

## Co-immunoprecipitation

For protein extraction, cells were washed with cold PBS (Corning) and lysed with IP lysis buffer (50 mM HEPES pH 7.4, 150 mM NaCl, 2 mM EDTA, 1% (v/v) NP-40, 30 mM sodium pyrophosphate, 2 mM sodium orthovanadate and 0.2% protease inhibitors in dH$_2$O) on ice for 30 min. Then, lysates were centrifuged at maximum speed for 30 min and the supernatant was collected. Protein concentration was determined using a BCA assay (ThermoScientific).

Protein G Mag Sepharose beads (Cytiva) were washed in lysis buffer and incubated for 1 h on rotation with 2 μg/mL of anti-FLAG antibody (Merck #F3165). Following incubation, beads were washed with lysis buffer and incubated with the lysates overnight at 4 °C. Then, beads were washed three times with washing buffer (50 mM HEPES pH 7.4, 150 mM NaCl, 2 mM EDTA and 0.1% NP-40), followed by one additional NP-40. Finally, beads were eluted in 30 μL loading buffer and incubated for 10 min at 95 °C. The eluted proteins were loaded onto an SDS-PAGE for western blotting.

## Immunofluorescence staining and imaging

For immunofluorescence staining, cells were fixed in 4% formaldehyde solution for 15 min and permeabilized with 0.05% Triton X-100 for 10 min. Samples were then blocked with 10% goat serum for 1 h at room temperature and incubated overnight at 4 °C with the primary antibodies anti-FGFR1 (Abcam #ab76464) and anti-α-tubulin (Merck #F6199). After primary antibody incubation, samples were washed and incubated for 1 h at 37 °C with the fluorophore-conjugated secondary antibodies anti-mouse Alexa Fluor 488 (Invitrogen #A21202) and anti-rabbit Alexa Fluor 555 (Invitrogen #A21430). Finally, samples were mounted with Prolong Gold Antifade Reagent with DAPI (Invitrogen), and images were captured using a Leica SP5 inverted confocal microscope.

## CRISPR/Cas9 genome editing

The coding sequences for single-guide RNAs (sgRNAs) designed to target specific *FGFR1* loci have been cloned into the PX458 vector (Addgene #48138) as described in (Ran et al, 2013) and delivered through lipofection in HOG cells together with specific synthetic donor single-strand DNAs (Alt-R™ HDR Donor Oligos, Integrated DNA Technologies). Donor templates were designed to introduce additional silent point mutations for PAM silencing and restriction enzyme-based screening (Appendix Fig. S4). Cells were sorted in multiwell96 plates as single cells per well and colonies were

identified after two weeks. To evaluate sgRNAs efficacy, test based on cleavage with T7 endonuclease (New England Biolabs) have been carried out prior to experiment. Clones have been grown and screened through PCR followed by cleavage with specific restriction enzymes and editing has been confirmed by Sanger sequencing. Primer sequences for PCR reactions, sgRNA and donor template sequences are specified in Appendix Tables S2–S4.

## Clonogenic assays

For 2D clonogenic assays, cells were seeded in six-well plates at a density of $10^3$ cells per well (HOG) or $10^2$ cells per well (MO3.13) in fresh medium and incubated for 2 weeks. After incubation, colonies were fixed with methanol for 10 min and stained with 0.1% (w/v) crystal violet in $dH_2O$ for 20 min. Plates were then washed with $dH_2O$ to remove the excess of stain crystal and allowed to dry. Stained plates were scanned, and colonies were quantified using ImageJ software.

For 3D colony formation experiments in Matrigel, 100 µL of Matrigel (Corning) was added to 1 cm² wells and led solidify for 15 min at 37°. Subsequently, $1 \times 10^4$ cells were seeded into each well and incubated for 1 week at 37 °C in a 5% $CO_2$ atmosphere. Images were captured using a Carl Zeiss™ Axio Vert.A1 microscope (Zeiss Zen software) and colony area was quantified for colonies with a minimum size of 100 µm² using ImageJ software. Each condition was plated in duplicate (two wells) and three independent fields were analyzed for each well.

## Quantitative RT-PCR

RNA was extracted from the cells using the RNeasy kit (Qiagen), and RNA concentration was determined using a Nanodrop One spectrophotometer. cDNA was synthesized using the SuperScript Reverse Transcriptase II kit (Invitrogen) and quantitative PCR reactions were performed using the PowerUp SYBR Green Master Mix (Applied Biosystems) on a LightCycler 480 thermocycler. The data was analyzed using the $2^{-\Delta\Delta Ct}$ method, calculating mean ± SEM using three replicates per sample and correcting for housekeeping genes (*HPRT1* for HEK293 cells and *ACTB* for HOG cells) expression values. Primer sequences have been specified in Appendix Table S5.

## 3D protein structure prediction

In the absence of a crystal structure of the full FGFR1-PLCγ protein complex in the Protein Data Bank, we employed AlphaFold3 (AF3, developed by DeepMind (Abramson et al, 2024)) to generate an approximate model of the complex. This predictive approach allowed us to explore the potential location of the binding interaction between PLCγ and the FGFR1 kinase domain, yielding high-confidence metrics for both the interfacial predicted modeling (ipTM = 0.45) and the predicted template modeling (pTM = 0.54), with pIDDT values above 70 for residues located at the interface.

The selected models were refined using the QuickPrep protocol in MOE2020 (Molecular Operating Environment (MOE), 2024). The AlphaFold3 model for FGFR1 (*Homo sapiens*, UniProt ID: P11362) shares 99.7% similarity with the crystallized human active form (PDB ID: 3GQI), achieving 77.3% overlap. Structural alignment of the model with the original pocket residues displayed

a strong superposition, with an average RMSD of 4.81 Å (Appendix Fig. S3). Overall, the resulting complex FGFR1-PLCγ predicted by AlphaFold3 agrees with previously reported experimental data by Hajicek and colleagues (Hajicek et al, 2019).

## RNA library construction and sequencing

RNA samples were collected in triplicates using the RNeasy kit (Qiagen) and integrity was evaluated using RNA 6000 Nano Assay on a Bioanalyzer 2100 (Agilent). Libraries were generated using the TruSeq® Stranded mRNA LT Sample Prep Kit (Illumina Inc., Rev.E, October 2013). Libraries were sequenced by the standard Illumina protocol to create raw sequence files (.fastq files), which underwent quality control analysis using FastQC. To avoid low-quality data negatively influencing downstream analysis, we trimmed the Readson the 3′-end and only used the first 51 bp from the 5′-end of each read for further analysis.

## Analysis of RNA-seq data

RNA-seq reads were mapped against the human reference genome (GRCh38) with STAR 2.7.8a (Dobin et al, 2013) using ENCODE parameters. Genes were quantified with RSEM 1.3.0 (Li and Dewey, 2011) using the Gencode v44 annotation. Genes with at least 1 count-per-million reads (cpm) in at least 3 samples were kept. Differential expression analysis was performed with limma R package v3.54.2 (Ritchie et al, 2015) using TMM normalization. The voom function (Law et al, 2014) was used to transform the count data into log2-counts per million (logCPM), estimate mean-variance relationship and to compute observation-level weights. These voom-transformed counts were used to fit the linear models. Volcano plots were generated using the ggplot R package. Gene ontology (GO) was performed using the clusterProfiler (version 3.0.4) R package, correcting for False Discovery Rate (FDR) through the Benjamini–Hochberg method. Cutoffs for significant differentially expressed genes were set at Log2FC < -0.7 or >0.7 and FDR ≤ 0.05. GO plots were generated through the ggplot2 (3.5.1) R package.

## Statistical analysis

Correlation analysis between groups has been performed through a two-sided Chi-square test with a confidence interval of 95%.

Data obtained from cellular assays and molecular analysis was tested using Analysis of Variance (ANOVA) or Kruskal–Wallis method (when normality was not assumed). For multiple comparisons, a post hoc pairwise comparisons analysis was conducted using the Tukey Honest Significant Difference (Tukey HSD) test for ANOVA or Dunn's test for Kruskal–Wallis test, to assess the significant differences between groups. All statistical tests were performed at an adjusted significance level of $P < 0.05$. The statistical analysis was performed using R (R Core team 2022, R version 4.3.3).

# Data availability

BioID-MS data has been deposited as a complete submission to the MassIVE repository (https://massive.ucsd.edu/ProteoSAFe/static/

massive.jsp) and assigned the accession number MSV000096690. The dataset is accessible at ftp://massive-ftp.ucsd.edu/v10/MSV000096690/. RNAseq data has been deposited in NCBI's Gene Expression Omnibus and is accessible through GEO Series accession number GSE286238.

The source data of this paper are collected in the following database record: biostudies:S-SCDT-10_1038-S44318-025-00600-3.

## Peer review information

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

## Acknowledgements

We would like to thank Prof. William Foulkes for the support along all stages of this work. Prof. Lopes-Guerrero, Prof. Querol Gutierrez and Dr Antonio Gentinella for the kind gift of the HOG cell line, MO3.13 cell line and the Streptavidin-HRP conjugate reagent, respectively. Dr Meritxell Rovira and Dr Cristina Muñoz for sharing antibodies and reagents. We would like to thank also Dr Tenzin Gayden for the help in the early stages of this conceptual work and the MNFC peer mentoring group. Dr Sidong Huang, Dr Ruth Rodriguez and Dr Marta Pineda for helpful discussion. Finally, we also acknowledge the Flow Cytometry and Bioimaging facilities of IDIBELL institute and Helena Martí for helping with figure making. BR is a Miguel Servet Fellow (CP21/00038) from the Carlos III Health Institute. JB was supported by a Juan de la Cierva fellowship (FJC2020-045392-I) from the Spanish National Plan for Scientific and Technical Research and Innovation. MF contract has been funded by Ministerio de Trabajo y Economía Social through Programa Investigo, grant number 2022-C23.I01.P03.S0020-0000209, funded by the European Union - Next Generation EU funds, Plan de Recuperación, Transformación y Resiliencia. HH benefited from a predoctoral fellow (PERIS, 2021-2024, with reference SLT017/20/000243) from the Government of Catalonia. AA is a Miguel Servet Fellow (CP23/00115) from the Carlos III Health Institute. IEE was received support from Fonds de recherche du Québec-Santé (FRQS) Doctoral Award and currently holds a Canadian Cancer Society Postdoctoral Research Fellowship Award (#708442). JFC holds the Tier-1 Canada Research Chair in Cellular Signalling and Cancer Metastasis and the Alain Fontaine Chair in Cancer from the IRCM Foundation. This work has been supported by a YIA award of the Alex's Lemonade Stand Foundation to BR, a Mia Neri YIA Award to BR, a La Caixa Junior Leader Grant (LCF/BQ/PI19/11690009) to BR, a Consolidator Grant (CNS2023-144251) to BR, funded by MCIN/AEI and by "European Union NextGenerationEU/PRTR". In addition, this project has been also funded by Instituto de Salud Carlos III through the project PMP22/00064, implemented under the NextGenerationEU funds, which finance the actions of the Mecanismo de Recuperación y Resiliencia. It has also received partial funding from a Project Grant from the Canadian Institute Health Research (CIHR, PJT-178083) to JFC, and the project PID2022-136344OA-I00 to AA funded by the Spanish Ministry of Science and Innovation (MCIN/AEI), the CERCA Program/Generalitat de Catalunya, and FEDER funds/European Regional Development Fund (ERDF) – a way to Build Europe.

## Author contributions

**Jacopo Boni**: Conceptualization; Data curation; Formal analysis; Investigation; Visualization; Methodology; Writing—original draft; Writing—review and editing. **Míriam Fernández-González**: Data curation; Formal analysis; Investigation; Visualization; Methodology; Writing—original draft. **HyeRim Han**: Investigation; Methodology; Writing—review and editing. **Carla Roca**: Data curation; Formal analysis; Writing—review and editing. **Cassandra J Wong**: Investigation; Methodology. **Cristina Rioja**: Formal analysis; Project administration; Writing—review and editing. **Clara Nogué**: Investigation; Visualization; Methodology. **Leticia Manen-Freixa**: Formal analysis. **Jonathan Boulais**: Software; Formal analysis. **Endika Torres-Urtizberea**: Formal analysis. **Antonio Gomez**: Data curation; Software. **Martin Hasselblatt**: Investigation; Writing—review and editing. **Roger Estrada-Tejedor**: Supervision. **Albert A Antolin**: Software; Supervision. **Islam E Elkholi**: Data curation; Writing—review

and editing. **Nada Jabado**: Supervision. **Jean-Francois Côté**: Supervision. **Anne-Claude Gingras**: Supervision. **Barbara Rivera**: Conceptualization; Supervision; Funding acquisition; Investigation; Writing—original draft; Project administration; Writing—review and editing.

Source data underlying figure panels in this paper may have individual authorship assigned. Where available, figure panel/source data authorship is listed in the following database record: biostudies:S-SCDT-10_1038-S44318-025-00600-3.

## Disclosure and competing interests statement

AA is/was a consultant of DarwinHealth and has received funds from VIVAN Therapeutics and AtG Therapeutics.

# Expanded View Figures

**Figure EV1.  Changes in receptor activation, ERK signaling and phenotype in FGFR1-mutant HEK293 cell lines.**

(**A**) Western blot of the parental (empty-Flag) and the six Flp-In T-REx HEK293 cell lines showing expression and activation of FGFR1 receptor and downstream MAPK/ERK signaling pathway (total and phosphorylated ERK1/2), after 24 h of Tet induction. (**B**) Bar plot of relative amounts (quantifications of western blots from $n = 6$ independent experiments) of autophosphorylated WT and mutant Flag-FGFR1 protein levels (as in Fig. 2C) obtained from the six T-REx HEK293 cell lines, induced with doxycycline (24 h). Intrinsic autophosphorylation rates shown in the plot have been evaluated correcting phospho-tyrosines (Y653/654) outputs against each respective total Flag-FGFR1 protein amount. Fold change (FC) values, obtained normalizing against WT levels, are indicated. Error bars represent standard errors of the mean ± SEM values. Significant comparisons against WT values are indicated (Kruskal–Wallis test); *P* value: *$P = 0.0309$. (**C**) Western blot of the six cell lines used in the present study, showing expression and activation (phosphorylation of Y653/654 residues) of FGFR1, after 24 h of Tet induction and under serum starvation (0% FBS overnight) conditions. (**D**) Optical microscope captures of T-REx Flp-In HEK293 expressing WT and mutant FGFR1 proteins 24 h post Tet-induction. Scale bar: 200 µm. (**E**) Dot plot of GO enrichment analysis using the list of proteins found in Clusters 14-15 of the heatmap in Fig. 2H (R661P-specific + WT/R661P shared interactors) as input list. Significantly enriched (adj. *P* value ≤ 0.05, Benjamini–Hochberg method) have been represented. Size and color of dots represent numbers of matched genes and adjusted *P* value (*P*.adjust), respectively. GO Gene Ontology.

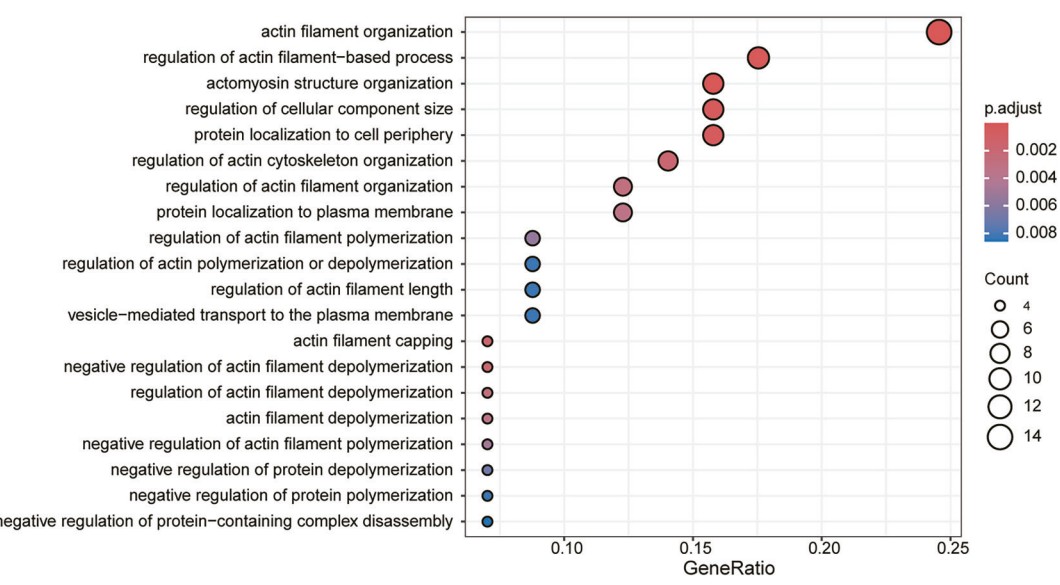

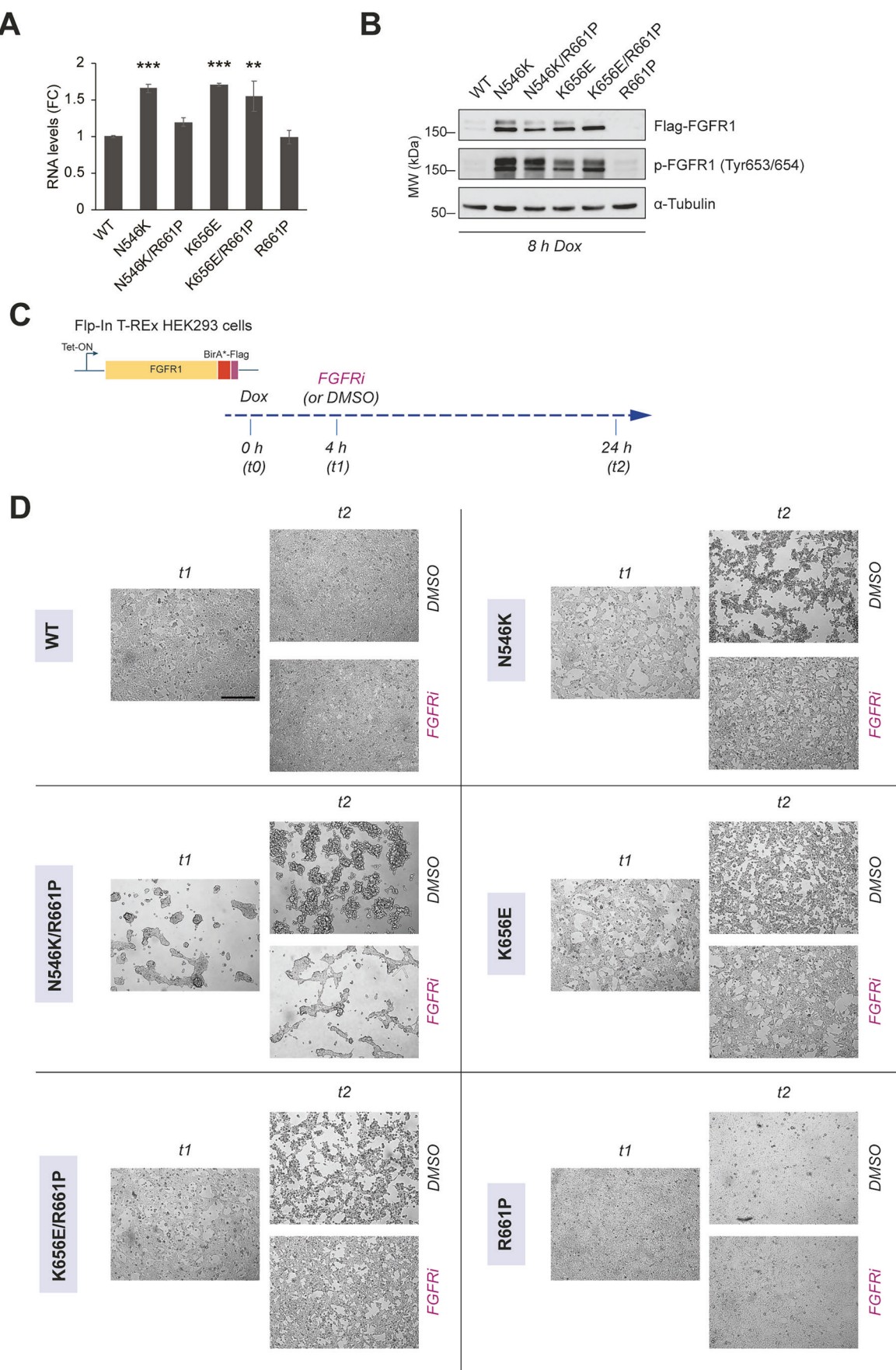

◀ **Figure EV2. N546K- and K656E-mediated protein accumulation and transformation are dependent on FGFR1 hyperphosphorylation.**

(A) FGFR1 RNA levels in WT and the five mutant T-REx Flp-In HEK293 cell lines, obtained by RT-qPCR. Data is represented by mean ± SEM from $n = 2$ independent experiments and significant differences against WT condition have been indicated (ANOVA test); $P$ values from left to right: ***$P = 0.00094$, ***$P = 0.00017$, **$P = 0.00731$. (B) Western blot assays revealing Flag-FGFR1 expression and autophosphorylation after 8 h of Tet-induction (Dox). (C) Experimental design for experiments with FGFR inhibitor. Cells were treated four hours after inducing Flag-FGFR1 expression either with DMSO or AZD4547 compound and lysates were collected at 24 h post-Tet induction. (D) Optical microscope captures of T-REx Flp-In HEK293 expressing WT and mutant FGFR1 proteins 4 h (treatment starting timepoint) and 24 h (end timepoint) post Tet-induction. For every condition, end-point pictures represent cells treated either with DMSO (upper panel) or FGFRi (bottom panel). Scale bar: 200 μm.

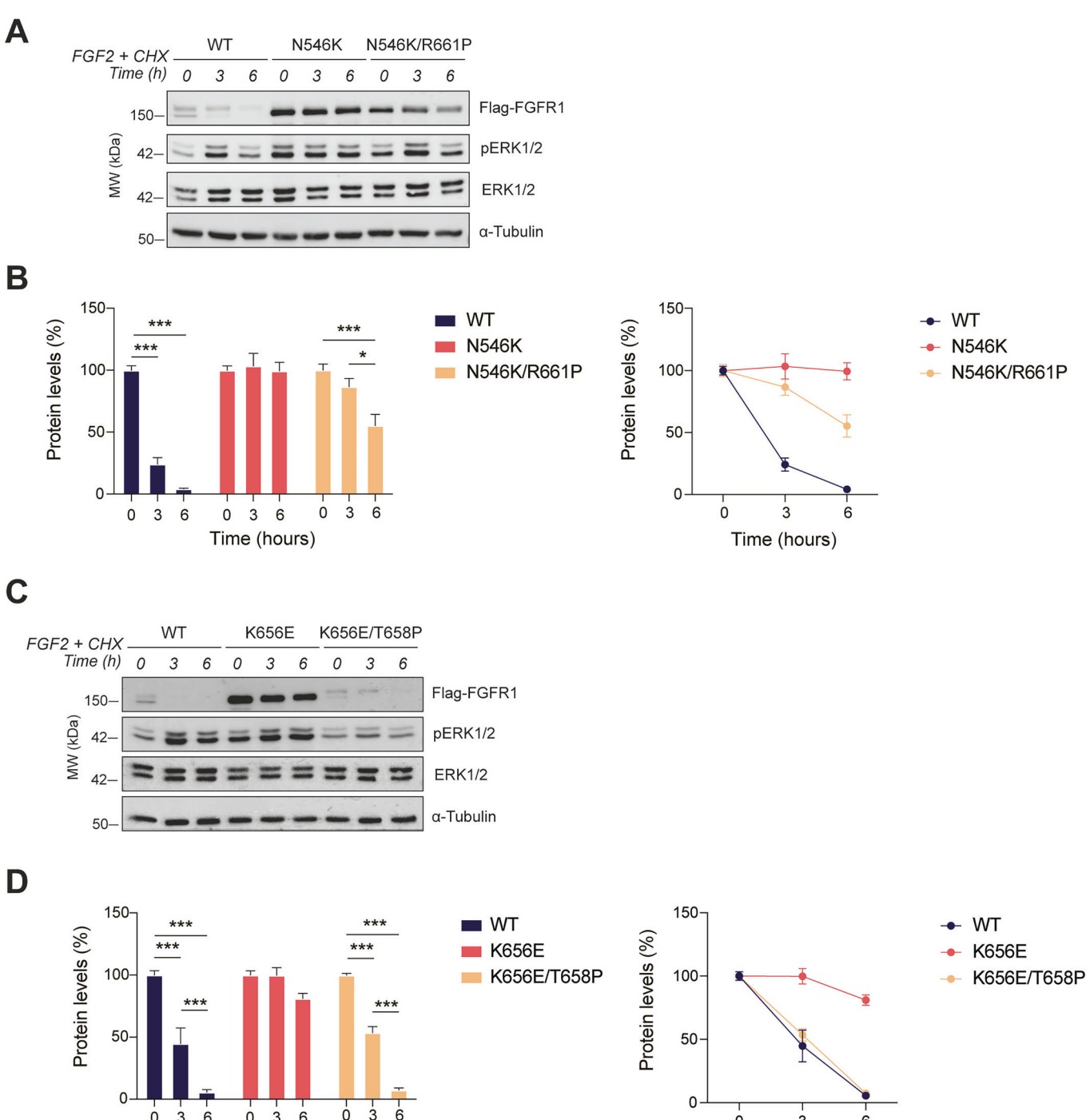

**Figure EV3.   Rescue of oncogenic FGFR1 proteins degradation mediated by secondary mutations.**

(A) Western blot analysis of a representative time-course experiment comparing WT FGFR1, N546K single and double mutants showing total Flag-FGFR1 as well as total and phospho-ERK1 protein levels at 0, 3 and 6 h post-treatment with FGF2 and cycloheximide (CHX). (B) Relative amounts of WT, N546K single and double (N546K/R661P) mutant FGFR1 protein obtained by quantifying western blots from n = 3 independent experiments. Values have been normalized (FC) against each relative 0 h reference values. In each plot, data is represented by mean ± SEM and significant variations in protein amounts against each specific reference values have been indicated in the bar plots (ANOVA test); $P$ values from left to right: ***$P$ < 1e-12, ***$P$ < 1e-12, ***$P$ = 0.00011, *$P$ = 0.0187. (C) Western blot analysis of a representative time-course experiment comparing WT FGFR1, K656E single mutant and K656E/T658P double mutant showing total Flag-FGFR1 as well as total and phospho-ERK protein levels at 0, 3 and 6 h post-treatment with FGF2 and cycloheximide (CHX). (D) Relative amounts of WT, K656E and K656E/T658P FGFR1 protein obtained by quantifying western blots from n = 3 independent experiments. Values have been normalized (FC) against each relative 0 h reference values. In each plot, data is represented by mean ± SEM and significant variations in protein amounts against each specific reference values have been indicated in the bar plots (ANOVA test); $P$ values from left to right: ***$P$ = 2.1e-08, ***$P$ < 1e-12, ***$P$ = 9.8e-05, ***$P$ = 2.3e-06, ***$P$ < 1e-12, ***$P$ = 2.7e-06.

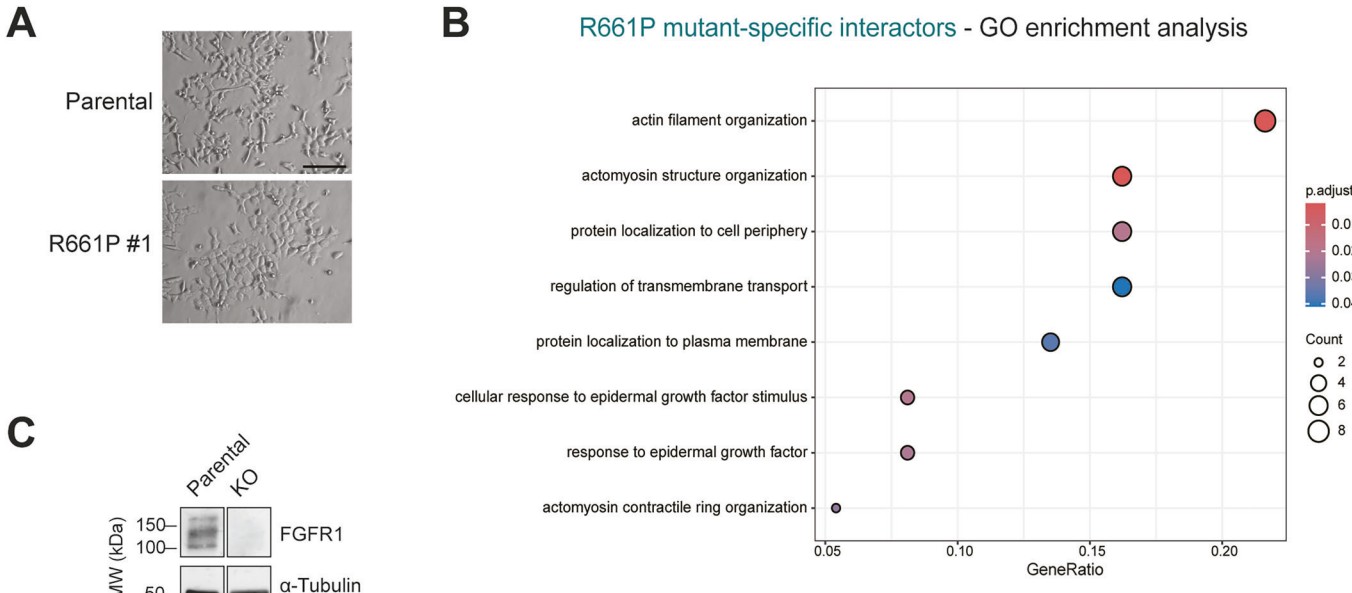

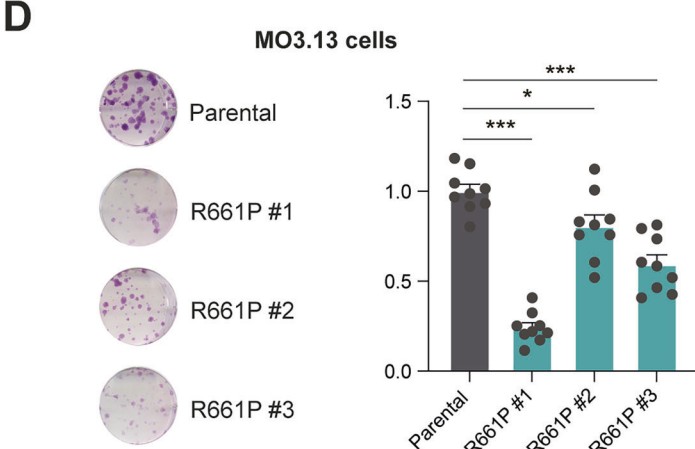

**Figure EV4.   FGFR1 R661P variant molecular and cellular associated phenotypes.**

(A) Optical microscope captures of Parental (upper panel) and R661P #1 (lower panel). Scale bar: 200 μm. (B) Dot plot of GO enrichment analysis using the list of preys forming Cluster 14 of the heatmap in Fig. 2H (BioID R661P-specific interactors). Size and color of dots represent numbers of matched genes and adjusted *P* value (p.adjust), respectively. (C) Western blot of FGFR1 protein confirming absence of FGFR1 expression in KO HOG cells, compared to Parental control. GO Gene Ontology. (D) Colony forming assays comparing CRISPR-edited, FGFR1-R661P MO3.13 clones with the parental cell line. Three homozygous R661P/R661P clones have been assayed. Microscope captures of representative wells for each condition have been included on the left. Quantification of $n \geq 3$ experiments have been plotted, normalized against the parental cell line. Replicates are represented by black dots. Data is represented by mean ± SEM and significant variations in protein amounts against each specific reference values have been indicated (ANOVA test); *P* values from left to right: ***$P = 4.3e-12$, *$P = 0.0306$, ***$P = 4.4e-06$.

