## [Peer Review File · The EMBO Journal]

Concurrence of *FGFR1* mutations modulates oncogenesis in glioneuronal tumors

Jacopo Boni, Míriam Fernández-González, HyeRim Han, Carla Roca, Cassandra Wong, Cristina Rioja, Clara Nogué, Leticia Manen-Freixa, Jonathan Boulais, Endika Torres-Urtizberea, Antonio Gomez, Martin Hasselblatt, Roger Estrada-Tejedor, Albert Antolin, Islam Elkholi, Nada Jabado, Jean-Francois Côté, Anne-Claude Gingras, and Barbara Rivera

Corresponding author: Barbara Rivera (brivera@idibell.cat)

Review Timeline:

Submission Date:	17th Jan 25
Editorial Decision:	15th Mar 25
Revision Received:	26th Jul 25
Editorial Decision:	29th Aug 25
Revision Received:	10th Sep 25
Accepted:	1st Oct 25

Editor: Daniel Klimmeck

Transaction Report:

Dear Dr Rivera,

Thank you again for the submission of your manuscript (EMBOJ-2025-120199) to The EMBO Journal. Please accept my apologies for the unusual delay with the peer-review of your manuscript at this time of the year. Your manuscript has been initially sent to three reviewers, however one reviewer got much delayed and in the end did not send us his-her report even after repeated chasers. We have received reports from the other two referees, which I enclose below, and now decided to proceed with our decision based on these comments.

As you will see from the experts' reports, the referees acknowledge the analysis and potential interest and value of your findings. However, they also express important issues regarding the completeness of your study and physiological relevance - generality of the results, which need to be addressed thoroughly to make them supportive of publication in the EMBO Journal. Further, the reviewers raise a number of issues related to the presentation of the findings, statistics applied and overall discussion of related literature, that would need to be conclusively addressed to achieve the level of robustness and clarity needed for The EMBO Journal.

Given the overall interest stated and broader angle of your findings, we are able to invite you to revise your manuscript experimentally to address the referees' comments. I need to stress though that we do require strong support from the referees on a revised version of the study in order to move on to publication of the work.

I would appreciate if you could contact me during the next weeks for exchange e.g. a video call to discuss your perspective on the comments and potential plan for revisions.

Please feel free to contact me if you have any questions or need further input on the referee comments.

When submitting your revised manuscript, please carefully review the instructions below.

Please feel free to approach me any time should you have additional questions related to this.

Thank you for the opportunity to consider your work for publication.

I look forward to your revision.

Best regards,

Daniel Klimmeck

Daniel Klimmeck, PhD
Senior Editor
The EMBO Journal

Instruction for the preparation of your revised manuscript:

- 1) a .docx formatted version of the manuscript text (including legends for main figures, EV figures and tables). Please make sure that the changes are highlighted to be clearly visible.
- 2) individual production quality figure files as .eps, .tif, .jpg (one file per figure).
- 3) a .docx formatted letter INCLUDING the reviewers' reports and your detailed point-by-point response to their comments. As part of the EMBO Press transparent editorial process, the point-by-point response is part of the Review Process File (RPF), which will be published alongside your paper.
- 4) a complete author checklist, which you can download from our author guidelines (<https://wol-prod-cdn.literatumonline.com/pb->

assets/embo-site/Author Checklist%20-%20EMBO%20J-1561436015657.xlsx). Please insert information in the checklist that is also reflected in the manuscript. The completed author checklist will also be part of the RPF.

6) It is mandatory to include a 'Data Availability' section after the Materials and Methods. Before submitting your revision, primary datasets produced in this study need to be deposited in an appropriate public database, and the accession numbers and database listed under 'Data Availability'. Please remember to provide a reviewer password if the datasets are not yet public (see <https://www.embopress.org/page/journal/14602075/authorguide#datadeposition>).

7) Our journal encourages inclusion of *data citations in the reference list* to directly cite datasets that were re-used and obtained from public databases. Data citations in the article text are distinct from normal bibliographical citations and should directly link to the database records from which the data can be accessed. In the main text, data citations are formatted as follows: "Data ref: Smith et al, 2001" or "Data ref: NCBI Sequence Read Archive PRJNA342805, 2017". In the Reference list, data citations must be labeled with "[DATASET]". A data reference must provide the database name, accession number/identifiers and a resolvable link to the landing page from which the data can be accessed at the end of the reference. Further instructions are available at .

8) At EMBO Press we ask authors to provide source data for the main and EV figures. Our source data coordinator will contact you to discuss which figure panels we would need source data for and will also provide you with helpful tips on how to upload and organize the files.

Numerical data can be provided as individual .xls or .csv files (including a tab describing the data). For 'blots' or microscopy, uncropped images should be submitted (using a zip archive or a single pdf per main figure if multiple images need to be supplied for one panel). Additional information on source data and instruction on how to label the files are available at .

9) We replaced Supplementary Information with Expanded View (EV) Figures and Tables that are collapsible/expandable online (see examples in <https://www.embopress.org/doi/10.15252/emj.201695874>). A maximum of 5 EV Figures can be typeset. EV Figures should be cited as 'Figure EV1, Figure EV2' etc. in the text and their respective legends should be included in the main text after the legends of regular figures.

11) For data quantification: please specify the name of the statistical test used to generate error bars and P values, the number (n) of independent experiments (specify technical or biological replicates) underlying each data point and the test used to calculate p-values in each figure legend. The figure legends should contain a basic description of n, P and the test applied. Graphs must include a description of the bars and the error bars (s.d., s.e.m.).

We realize that it is difficult to revise to a specific deadline. In the interest of protecting the conceptual advance provided by the work, we recommend a revision within 3 months (13th Jun 2025). Please discuss the revision progress ahead of this time with the editor if you require more time to complete the revisions.

Referee #1:

This manuscript by Boni et al., investigates how a few FGFR1 mutations, particularly the co-occurrence of common hotspot variants (N546K, K656E) with a secondary allele (R661P), contribute to low-grade glioneuronal tumorigenesis. The authors performed public database analysis to establish the prevalence of FGFR1 hotspots (alone vs. in combination with secondary mutations) in brain tumors and used proximity labeling to map and compare proximal interactomes of wild-type FGFR1, single mutants (N546K, K656E, R661P), and double mutants (N546K/R661P, K656E/R661P) in Flp-In T-REx HEK293 cells. They detected differences in how the various mutants evade or undergo lysosomal degradation. Additionally using CRISPR/Cas9 gene editing to introduce the FGFR1 mutations (single or double) in HOG oligodendrogloma cells, the authors showed that R661P alone severely impairs colony formation, whereas the double mutants (hotspot + R661P) partially rescue proliferative or "stem-like" behavior. Finally the authors propose a modulatory mechanism whereby R661P can either impair or "rescue" FGFR1 signaling capacity in oligodendroglial contexts, explaining why multiple mutations in FGFR1 might be positively selected in certain low-grade brain tumors. Overall, this is a very well designed and executed article and strong candidate to be published once the remaining questions are resolved.

Major Criticisms

- The authors primarily rely on HOG cells to examine the functional consequences of FGFR1 mutations. HOG is a useful but immortalized line that may not perfectly mimic primary tumor biology or patient-derived cultures. Therefore validating the key findings in additional relevant models (another oligodendroglial line or patient-derived cultures) could increase the confidence that the R661P effects and double-mutant phenotypes are robust across multiple systems.
- Although the BioID-MS approach pinpoints significant changes in proximal interactors (e.g., PLC γ), the manuscript only minimally characterizes broader signaling outputs such as the MAPK/ERK or PI3K/AKT axes. Demonstrating how these FGFR1 mutations impact classical FGFR-dependent pathways would strengthen mechanistic inferences and clarify how partial "rescue" by R661P is actually manifested at the signaling level.
- The paper emphasizes receptor accumulation and evasion of lysosomal degradation by oncogenic mutants but only briefly addresses whether intrinsic kinase activity is also changed under each mutational context. Clarifying if the primary oncogenic mechanism is increased half-life or also hyperactivated enzymatic function would provide a more complete view of how these FGFR1 mutations drive tumorigenesis.
- The authors suggest that stability of N546K/K656E FGFR1 could reduce inhibitor efficacy, yet no drug response assays directly demonstrate how these mutants respond to FGFR-targeted therapies. Without empirical data, discussions of therapy resistance or "optimal therapeutic approaches" remain somewhat speculative. Showing actual sensitivity or resistance in CRISPR lines would be more conclusive. Alternatively the authors could tone down their claim.

Referee #3:

In this manuscript, the authors explore potential oncogenic mechanisms of specific FGFR1 variants. They find that FGFR1 N546K and K656E variants are more common in brain tumours than other anatomical locations and note that only brain tumours appear to demonstrate multiple hits. By investigating the proximal interactome of WT, N546K, and K656E FGFR1, they highlight functional changes associated with oncogenic variants. This latter point is extended to double-mutant models incorporating either N546K or K656E and an additional R661P variant, with evidence these combinations have their own additional functional changes. Noting that the mutant models possess greater levels of protein accumulation, they present evidence via lysosomal degradation assay that the oncogenic N546K/K656E variants escape lysosomal protein degradation, a process that appears to be partially rescued in the presence of the R661P additional variant. In the latter sections of the manuscript, the authors shift focus towards R661P, investigating its modulatory effects.

The data presented around N546K and K656E is interesting, of broad interest, and appears novel. The manuscript is technically

sound as far as I can see. There is a lot of data included, and the manuscript feels rather like two separate studies: one on N456K/K656E and another specifically focusing on R661P. My primary queries relate to the work on the latter variant.

Major comments:

1. I struggle to see the broader applicability of the R661P work. My understanding from the literature and my own experience with sequencing LGGNTs is that R661P is not a frequently identified variant, primarily reported in DNETs from three members of one family in the corresponding authors previous work (as well as a singular case in the present study). The focus on this variant over the more frequent in cis variants, and its use as the main multi-hit model, seems limiting with regards to the generalisability of the data across secondary mutations. As such, I'm curious as to why the authors have chosen to focus on this additional variant rather than the more frequent additional hits identified in their initial GENIE search? (Appendix Figure S1).
2. I'm not sure I fully comprehend the logic underlying the authors argument in the discussion that the compensatory effects of R661P prime cells towards acquiring activating mutations. This feels like a major overstatement beyond what the data demonstrates.
3. Have the authors looked at whether tumours with multiple FGFR1 alterations in cis behave clinically differently compared to those with only a single variant? If additional variants can have modulatory effects on FGFR1 activity, as hypothesised in the manuscript, this may be visible in the tumour behaviour if a suitably sized cohort is available.

Specific additional comments.

1. At the end of the discussion: "The present study provides evidence of novel oncogenic mechanisms and modulatory effects derived from multiple FGFR1 alterations that need to be considered when choosing treatment options..." It would be useful if the authors highlighted specifics for how these alterations and their consideration can impact current treatment options.
2. On page 15, on the third line beneath "Generation of stable cell lines...", should "PMID: 11076863" be reformatted as a formal reference?
3. Figure 1 would be improved by adding n numbers to the figure for each of the percentages shown (e.g., "n = ..." next to the percentages on top of the bars for single/multiple variants. Likewise, n numbers in the legends for brain/others and low/high-grade categories).
5. Appendix Figure S1. It's unclear which of the A/B panels correspond to N546K and K656E. Please label the panels more clearly with the hotspot they represent at the top of each panel.
6. As a suggestion, it may be useful to include a histogram in the appendix showing the distribution of cases versus number of total variants per case, subdivided by N546K/K656E. Comparing the number of secondary variants in Appendix Figure S1 against the number of double/multiple samples in Table 2 it appears as if the majority may have multiple hits rather than just double?
7. Page 3, line 18 in the Introduction, dysembryoplastic is miss-spelled as "dysembryioplactic".

Please find below a point-by-point response to reviewers followed by a list of changes in figure position with respect to the first submitted version of the manuscript.

REFEREE #1:

This manuscript by Boni et al., investigates how a few FGFR1 mutations, particularly the co-occurrence of common hotspot variants (N546K, K656E) with a secondary allele (R661P), contribute to low-grade glioneuronal tumorigenesis. The authors performed public database analysis to establish the prevalence of FGFR1 hotspots (alone vs. in combination with secondary mutations) in brain tumors and used proximity labeling to map and compare proximal interactomes of wild-type FGFR1, single mutants (N546K, K656E, R661P), and double mutants (N546K/R661P, K656E/R661P) in Flp-In T-REx HEK293 cells. They detected differences in how the various mutants evade or undergo lysosomal degradation. Additionally using CRISPR/Cas9 gene editing to introduce the FGFR1 mutations (single or double) in HOG oligodendroglioma cells, the authors showed that R661P alone severely impairs colony formation, whereas the double mutants (hotspot + R661P) partially rescue proliferative or "stem-like" behavior. Finally, the authors propose a modulatory mechanism whereby R661P can either impair or "rescue" FGFR1 signaling capacity in oligodendroglial contexts, explaining why multiple mutations in FGFR1 might be positively selected in certain low-grade brain tumors. **Overall, this is a very well designed and executed article and strong candidate to be published once the remaining questions are resolved.**

We thank the reviewer for recognizing the value of our work and we address her/his questions below.

- The authors primarily rely on HOG cells to examine the functional consequences of FGFR1 mutations. HOG is a useful but immortalized line that may not perfectly mimic primary tumor biology or patient-derived cultures. Therefore, validating the key findings in additional relevant models (another oligodendroglial line or patient-derived cultures) could increase the confidence that the R661P effects and double-mutant phenotypes are robust across multiple systems.

We agree with the reviewer that crucial results, especially the proliferative deficiency associated with R661P, would be more robust if validated in another model. The MO3.13 cell line is a human-human hybrid cell line with oligodendrocytic features (PMID:7707048). Like HOG cells, MO3.13 cells have been widely used in literature as a model of immature oligodendrocyte, to study oligodendroglial cell physiological and pathological processes (examples: PMID: 27638607; PMID: 32608563; PMID: 37184766). Thus, we used the MO3.13 cell line, to validate results obtained with FGFR1-R661P HOG cells. We generated three independent FGFR1-R661P MO3.13 clones through CRISPR-Cas9. Considering that the MO3.13 cell line is diploid for FGFR1 gene, we isolated clones that had introduced the mutation in homozygosis (R661P/R661P), to be able to precisely define the phenotype associated with the variant. Three independent homozygous R661P clones have been generated from CRISPR knock-in experiments, following the same approach used to isolate R661P HOG clones. Mutated genomic DNA sequences have been included in the manuscript and added to the Supplementary Figure S4B in the Appendix section. In clonogenic assays, R661P MO3.13 clones showed impaired clonogenic potential, compared to the parental cell line, validating the phenotype observed in HOG R661P. This data has been included in the manuscript as Fig. EV4D. The results have been elucidated in the text at page 11, lines 354-360.

- Although the BioID-MS approach pinpoints significant changes in proximal interactors (e.g., PLC γ), the manuscript only minimally characterizes broader signaling outputs such as the MAPK/ERK or PI3K/AKT axes. Demonstrating how these FGFR1 mutations impact classical FGFR-dependent pathways would strengthen mechanistic inferences and clarify how partial "rescue" by R661P is actually manifested at the signaling level.

-We thank the reviewer for raising this important point. We addressed this question by investigating the activation of the MAPK/ERK pathway mediated by FGFR1 single and double mutant receptors and how modulation exerted by R661P variant affects this activation. Therefore, we have assessed ERK1/2 activation by looking at phosphorylated (T202/Y204 residues) ERK1/2 protein levels in different settings:

-First, we have assessed ERK1/2 activation in our panel of HEK293 cell lines expressing WT and mutant FGFR1 proteins in standard conditions. As expected, activation of MAPK/ERK pathway was found increased in cells expressing oncogenic mutant FGFR1 (N546K and K656E). This plot has been included in the manuscript as Fig. EV1A and results illustrated in the text at page 6, lines 189-192: *"cells expressing the hotspot mutations N546K and K656E appeared with increased phosphorylated FGFR1 (p-FGFR1) levels and enhanced activation of downstream MAPK/ERK pathway, compared to the WT condition"*.

-We also screened our T-REx HEK293 cell models for MAPK/ERK signaling in response to the FGFR inhibitor AZD4547 (see answer to the next point), showing that inhibition of receptor autophosphorylation negatively affected ERK activation (in particular, in cells expressing N546K and K656E activating mutations), reverting pathway hyperactivation. These results are shown in the new plot included in the manuscript as Fig. 3E, and the effects on ERK activation have been stated at page 8, lines 248-249: *"The consequences of this regression were also seen in the activation of downstream MAPK/ERK signaling pathway"*.

-To determine whether modulatory effects exerted by secondary mutations on hotspot mutant oncogenic potential were reflected at the downstream MAPK/ERK signaling level, we re-run western blots using samples from the same stability experiments, described in the Fig. 4 of the old version, and assessed total and phosphorylated ERK levels. Notably, while remaining stable in single mutants N546K and K656E, ERK activation was reduced in double mutants at 6 hours post-induced degradation, reflecting differences in degradation rates. This observation can be found in the new version of the manuscript at page 9, lines 281-283: *"This modulatory effect played by the secondary mutation also affected downstream pathway activation, as ERK1/2 phosphorylation was also reduced at 6h in double mutant-expressing cells"*, and new plot with ERK/p-ERK levels are represented in Fig. 4C (K656E mutations) and Fig. EV3A (N546K).

-Finally, these last results showing changes in the activation of MAPK pathways as a result of the modulatory activity exerted by the R661P variant on oncogenic mutant N546K and K656E was also validated using a different double mutant (K656E/T658P) by experiments included in the revised version of the manuscript in Fig. EV 3C-D and described in lines 287-289: *"Similar to K656E/R661P, double mutant K656E/T658P showed reduced protein stability and attenuated activation of MAPK/ERK signaling compared to K656E"*.

- The paper emphasizes receptor accumulation and evasion of lysosomal degradation by oncogenic mutants but only briefly addresses whether intrinsic kinase activity is also changed under each mutational context. Clarifying if the primary oncogenic mechanism is increased

half-life or **also** hyperactivated enzymatic function would provide a more complete view of how these FGFR1 mutations drive tumorigenesis.

We agree with the reviewer that this aspect is crucial and that the link between tyrosine kinase activity and protein accumulation should be better clarified. We tackled this question through two strategies. First, we performed a systematic time-course experiment, inducing expression of Wt and all the five mutant FGFR1 proteins, assessing levels of total Flag-FGFR1 and phospho-Tyr at each time point. On the other hand, we treated FGFR1-expressing stable HEK293 cell lines with the FGFR inhibitor AZD4547, monitoring changes in total phosphorylated forms of FGFR1 proteins.

Results obtained through these experiments (illustrated below) were very revealing and allowed us to better understand how receptor activation and increased cellular stability were linked, which is the reason why we decided to include them in the main Figure 3 (3D and 3E, respectively), and in the manuscript at the end of the third section and the beginning of the fourth (pages 7-8, lines 226-253), rewiring the previous version and adding description of the new data.

Time-course (Fig. 3D): Reproducing what we already showed in previous Fig. 3D (now moved to Fig EV2B), while WT protein is barely detected, expression oncogenic mutants was already observed at 8 hours, indicating that WT protein is already targeted for degradation at early stages, through mechanisms that are evaded by oncomutant FGFR1 receptors. Notably, the highest peak of WT protein accumulation is reached at 12 hours, with levels that are decreased at 24 hours. This finding indicates that, even without inducing receptor internalization and degradation, the WT protein is intrinsically degraded during all stages of its cellular life and, that at 24 hours, HEK293 cells have increased the rate of degradation, most likely as a response mechanism to massive expression/accumulation. On the contrary, oncogenic mutants maintain total levels stable (or even increased) at 24 hours, reinforcing our model that these mutations lead to escaped degradation. However, oncogenic mutants also show hyperphosphorylation at early stages during protein expression, with high-intensity p-Tyr bands that can be observed already at 8 hours. As we state in the new version of the manuscript, this observation strongly suggests that the two properties might be co-operating in mediating oncogenic phenotypes in tumor cells.

FGFR inhibitor (Fig. 3E): To address the question of which is the “primary” oncogenic mechanism, also a fair concern of the reviewer, we used the same approach designed to tackle the next point, regarding how FGFR1 mutants respond to FGFR inhibitors. We treated our cell lines with AZD4547 or DMSO and collected total lysates at 24 post-induction, blotting total Flag-FGFR1 and phospho-FGFR1 levels. As shown in figure 3E, treatment with the FGFRi was sufficient to abolish phosphorylation of Tyr653/654 of WT and mutant FGFR1 proteins. Strikingly, while un-phosphorylated WT FGFR1 protein shows total levels comparable to control (vehicle), all oncogenic mutants display WT-like outcome (band pattern and intensity) when treated with the inhibitor, which means a dramatic decrease in oncogenic protein accumulation, compared to control condition. This data indicates that the two acquired oncogenic properties, tyrosine kinase activity and stability are linked and that hyperphosphorylation is required for the escape of degradation processes by N546K and K656E FGFR1 proteins.

According to the new findings, we have also changed the title of the third section to: “FGFR1 oncogenic mutants rapidly undergo autophosphorylation and are highly stable in human cells” (lines 187-188). Moreover, since we explored more in detail mechanisms of degradation through lysosomes in the following fourth section, old Fig. 3E (BafA1 treatment) has been

moved to Fig. 4 as Fig. 4A. Finally, we have removed the old Fig. 3F, as the new time-course included as Fig. 3D is now much more informative (5 time points).

- The authors suggest that stability of N546K/K656E FGFR1 could reduce inhibitor efficacy, yet no drug response assays directly demonstrate how these mutants respond to FGFR-targeted therapies. Without empirical data, discussions of therapy resistance or "optimal therapeutic approaches" remain somewhat speculative. Showing actual sensitivity or resistance in CRISPR lines would be more conclusive. Alternatively, the authors could tone down their claim.

We thank the reviewer for the comment. As we anticipated in the previous point, we decided to address this important question by treating our cell models with a FGFR inhibitor and monitoring changes in i) protein amounts (HEK293), ii) autophosphorylation (HEK293), iii) cell phenotype (HEK293 and HOG).

We chose the inhibitor AZD45747, or Fexagratinib, widely used in biomedical research and currently ongoing already tested in recently completed clinical trials for FGFR-mutated cancers. This compound, like other FDA-approved FGFR inhibitors, inhibits FGFR receptors (FGFR1, FGFR2, FGFR3) with the highest affinity for FGFR1.

-Results with HEK293 have been elucidated in the previous point. Considering their relevance in helping to understand biochemical/molecular events caused by the mutations investigated in our study, they have been included in the manuscript. In particular, they provide key data that clarify the molecular link between high kinase activity and increased stability/accumulation played by the oncogenic mutations N546K and K656E.

-We also performed experiments with our FGFR1-mutant HOG cell lines; results are shown in the Figure below. Pilot MTT assays underlined no significant differences in drug-response between different types of mutations (oncogenic N546K and germline/damaging R661P), nor with the parental (WT) cell line. This data suggests that, while the compound can inhibit phosphorylation of overexpressed WT and mutated FGFR1 receptor in HEK293 cells, partially reversing the transformed phenotype, response in physiological models (oligodendrocyte and endogenous expression levels) may be more limited. Importantly, no significant differences were observed with FGFR1-KO HOG cells, indicating that non-specificity of the AZD45747 inhibitor (targeting also FGFR2 and FGFR3 receptors) is an important aspect to consider when interpreting drug-response results. This data is still preliminary and future work on inhibitors using different FGFR inhibitors (including emerging new FGFR1-specific compounds) is required to establish response to treatment and will be part of our ongoing future work. For these reasons we decided to include these last results (drug response in HOG cells) only in the rebuttal letter for reviewers' information.

Figure for reviewers removed

Following reviewers comment we have toned down our claim in the discussion: i) we deleted conclusions not fully supported by our data, especially considering new results included in the revised version; ii) we included high tyrosine activity as part of the oncogenic mechanisms (line 407) triggered by N546K and K656E mutations and iii) reformulated the text (lines 433-445), providing possible explanations for the limited results obtained so far with FGFR inhibitors, and suggesting that further studies with more specific FGFR1 inhibitors and FGFR1-targeting PROTACs might provide alternative options for future treatments.

REFEREE #3:

In this manuscript, the authors explore potential oncogenic mechanisms of specific FGFR1 variants. They find that FGFR1 N546K and K656E variants are more common in brain tumours than other anatomical locations and note that only brain tumours appear to demonstrate multiple hits. By investigating the proximal interactome of WT, N546K, and K656E FGFR1, they highlight functional changes associated with oncogenic variants. This latter point is extended to double-mutant models incorporating either N546K or K656E and an additional R661P variant, with evidence these combinations have their own additional functional changes. Noting that the mutant models possess greater levels of protein accumulation, they present evidence via lysosomal degradation assay that the oncogenic N546K/K656E variants escape lysosomal protein degradation, a process that appears to be partially rescued in the presence of the R661P additional variant. In the latter sections of the manuscript, the authors shift focus towards R661P, investigating its modulatory effects.

The data presented around N546K and K656E is interesting, of broad interest, and appears novel. The manuscript is technically sound as far as I can see. There is a lot of data included, and the manuscript feels rather like two separate studies: one on N456K/K656E and another specifically focusing on R661P. My primary queries relate to the work on the latter variant.

We thank the reviewer for valuing our work in its novelty, soundness and interest to the field and address the specific comments below.

- I struggle to see the broader applicability of the R661P work. My understanding from the literature and my own experience with sequencing LGGNTs is that R661P is not a frequently identified variant, primarily reported in DNETs from three members of one family in the corresponding authors previous work (as well as a singular case in the present study). The focus on this variant over the more frequent in cis variants, and its use as the main multi-hit model, seems limiting with regards to the generalisability of the data across secondary mutations. As such, I'm curious as to why the authors have chosen to focus on this additional variant rather than the more frequent additional hits identified in their initial GENIE search? (Appendix Figure S1).

There are several reasons to justify our choice of variant R661P as the one to model. The first and more important one is that we can solidly establish the R661P can be acquired first followed by a hotspot somatic. Notably, R661P has been seen both in hereditary as well as in sporadic cases (with a VAF compatible with a somatic origin) meaning the combination of both variants is selected forth in both scenarios regardless of the order of acquisition. Moreover, we know R661P can appear both with N546K and K656E, so it allowed us to look for general effects over each of the recurrent alleles. Apart from these facts regarding the combination of alleles that allow us derived conclusions, there is already much information we can extract from the R661P being a germline allele. It perfectly segregates with susceptibility to the disease (DNETs) but it differs from any other congenital phenotype of germline pathogenic variants in *FGFR1*, supporting the specificity in the changes affecting the glial cells. Since the carriers have otherwise a completely normal phenotype, we expected the functions of R661P would not differ much from a WT *FGFR1* which is in fact concordant with most of the in vitro results and interactome profiles we have obtained.

Despite all these reasons, we understand the reviewer's concerns regarding how extensible the results of the R661P can be to other mutants. Thus, we have also modeled a double K656E/T658P mutant into the Flp-In T-REx HEK293 system. Results of these experiments can be found in the new version of the manuscript in Fig. EV3C-D, elucidated at page 9, lines 281-290. As it can be seen in the figure, the double mutants K656E/T658P fully recapitulate the rescue in onco-protein degradation, the upper band of *FGFR1* as well as the decrease in phosphorylation of ERK, thus less activation of MAPK pathway, compared to the K656E oncogenic mutant.

-I'm not sure I fully comprehend the logic underlying the authors argument in the discussion that the compensatory effects of R661P prime cells towards acquiring activating mutations. This feels like a major overstatement beyond what the data demonstrates.

We thank the reviewer for this comment. In our hypothesis working model the R661P poses a growth constraint that might create a selective pressure in the glia cells to acquire the oncogenic mutations, but we agree with the reviewer that the data does not fully prove this, and we got carried away by our enthusiasm as geneticists and the novel susceptibility mechanisms that this event would mean. We have deleted the statement and reformulated the sentences that now reads as follows (lines 470-482).

“FGFR1 is expressed in glial and neural precursors and plays essential functions during brain tissue development and differentiation of neural cell lineages, including oligodendrocyte precursor cells (Furusho et al, 2011; Grabiec et al, 2016; Yoon et al., 2004). In this context, both the oligodendrocyte lineage background and the absence of WT allele expression in R661P HOG cells would contribute to reveal the phenotypic defects caused by the mutant receptor. Importantly, we showed here that double mutant K656E/R661P and N546K/R661P rescue both stemness/proliferative capacity and expression levels of development-related genes, increasing the repertoire of mutation specific features that characterize FGFR1 mutants. The R661P mutation in combination with the K656E is also found in one case (likely of sporadic origin

based on their variant allele frequency) from the GENIE cohort, arguing that the combination of these two events perpetuates tumorigenesis. These results may be unveiling a novel model of oncogene-associated susceptibility, based on the need for “correction” of proper signaling and transcriptional programs, which ultimately predisposes toward selection of oncogenic events conferring driving capacity to the mutated clone. Sporadic cases harboring other missense mutations (as the double mutant K656E/T658P) might share common mechanisms with the ones uncovered in the present work, thus enlightening why DNETs and other therapy naïve gliomas with a quiet genome acquire additional hits involving the FGFR1 gene”.

-Have the authors looked at whether tumours with multiple FGFR1 alterations in cis behave clinically differently compared to those with only a single variant? If additional variants can have modulatory effects on FGFR1 activity, as hypothesised in the manuscript, this may be visible in the tumour behaviour if a suitably sized cohort is available.

This is a tremendously interesting point that we have aimed to tackle several times through our investigations. Unfortunately, there is not enough published data with complete clinical information regarding outcome in a large enough series of double mutants. Even in the very recent work from Dr Bennett et al, PMID 40335748 where they look at clinical outcome of several FGFR1 mutants in pediatric glioma, the number of double mutants with clinical follow up is limited precluding statistical analysis. This is an on-going goal of our team that is collecting a large database of double *FGFR1* mutants.

Minors:

-At the end of the discussion: "The present study provides evidence of novel oncogenic mechanisms and modulatory effects derived from multiple FGFR1 alterations that need to be considered when choosing treatment options..." It would be useful if the authors highlighted specifics for how these alterations and their consideration can impact current treatment options.

- We have reformulated the statement to: "*The present study provides a testimony of novel oncogenic mechanisms and modulatory effects derived from multiple FGFR1 alterations that will be informative when defining mutation-specific treatment options and might help to identify novel therapeutic approaches for these patients.*", (page15, lines 498-501).

Moreover, as discussed in the discussion section, the increased stability of the mutants might open the door to considering PROTACs and drug discovery approaches aimed at identifying mutation-specific inhibitors might be promising paths to follow.

-On page 15, on the third line beneath "Generation of stable cell lines...", should "PMID: 11076863" be reformatted as a formal reference?

- We have corrected the mistake and added the reference.

-Figure 1 would be improved by adding n numbers to the figure for each of the percentages shown (e.g., "n = ..." next to the percentages on top of the bars for single/multiple variants. Likewise, n numbers in the legends for brain/others and low/high-grade categories).

- Following reviewer's suggestion, numbers have been added to Figure 1.

-Appendix Figure S1. It's unclear which of the A/B panels correspond to N546K and K656E. Please label the panels more clearly with the hotspot they represent at the top of each panel.

-We sincerely thank the reviewer for noticing the missing information. In fact, A and B panels did not correspond to N546K and K656E, respectively, but to the distribution of double mutants in all brain tumors (now left panel) and in low-grade brain tumors (right panel).

-As a suggestion, it may be useful to include a histogram in the appendix showing the distribution of cases versus number of total variants per case, subdivided by N546K/K656E. Comparing the number of secondary variants in Appendix Figure S1 against the number of double/multiple samples in Table 2 it appears as if the majority may have multiple hits rather than just double?

Table 2 shows the number of cases per mutation category found in GENIE. The vast majority of cases with a secondary mutation are annotated with two hits only, the hotspot plus another one, although some cases do present more than 2 variants, but these are scarce (This can also be seen in our discovery paper cases 9 -with 4 nucleotide changes in a row- and case 10, PMID: 26920151). Given that double mutants are far more common, we have compiled both double and multiple mutants as the same category, referring to those samples with a hotspot variant plus at least one other. Yet we have included a note in the table legend with the number of mutants that are specifically annotated as more than two.

-Page 3, line 18 in the Introduction, dysembryoplastic is miss-spelled as "dysembryioplastic".

We have corrected the misspelled word.

Changes applied to previous version of manuscript Figures:

Below we include a list of all changes in figures from first submission to the revised version as well as a list of new figures.

3A – Total Flag-FGFR1 plot only. Autophosphorylation plot moved to EV Figures (now EV1B)

3D – Moved to EV Figures (now EV2B)

3E – Moved to Figure 4 (now 4A)

3F – Removed

New figure 3D: new time course experiments

New Figure 3E: FGFR inhibitor experiments

Fig. 4A – now Fig. 4B

Fig. 4B-C – now Fig. C-D; ERK/pERK panels added

Fig. 4D-E – Moved to EV Figures (now Fig. EV3A-B); ERK/pERK panels added

New Figure EV1A: total and p-FGFR1 expression and ERK activation in WT/mutant FGFR1

New EV1B: autophosphorylation panel from old Fig. 3A

EV1A becomes EV1C

EV1B becomes EV1D

EV1C becomes EV1E

EV1D moved to another figure, now is EV2A

New EV2B: is old Fig. 3D

New EV2C: strategy of FGFR inhibitor experiments

New EV2D: microscope captures of FGFR inhibitor experiments

Fig. EV2A-B moved to Fig. 4 (4E-F)

New Fig. EV3A-B: old Fig. 4C-D + ERK activation panels

New Fig. EV3C-D: new experiments with T658P variant

Fig. EV3 becomes Fig. EV4 (EV4A-B-C are the same panels)

New Fig. EV4D: MO3.13 colony forming assays results

Appendix Figure S3 – Ponceau has been removed

Appendix Figure S4 is now S3

Appendix S5 is now S4; Sanger seq. Of FGFR1-R661P MO3.13 clones have been included in panel B

Dear Dr Rivera,

Thank you for submitting your revised manuscript (EMBOJ-2025-120199R) to The EMBO Journal, as well for your patience with our feedback at this time. Your amended study was sent back to the referees for their scientific reassessment, and we have received re-reports from both of them, which I enclose below. As you will see, the reviewers state that the work has been substantially enhanced by the revisions and they are now broadly in favour of publication.

Thus, we are pleased to inform you that your manuscript has been accepted in principle for publication in The EMBO Journal.

We now need you to take care of a number of issues related to formatting and data presentation as detailed below, which should be addressed at re-submission.

Please contact me at any time if you have additional questions related to below points.

Thank you for giving us the chance to consider your manuscript for The EMBO Journal. I look forward to your final revision.

Again, please contact me at any time if you need any help or have further questions.

Best regards,

Daniel Klimmeck

>> Author Contributions: Remove the author contributions information from the manuscript text. Note that CRediT has replaced the traditional author contributions section as of now because it offers a systematic machine-readable author contributions format that allows for more effective research assessment. and use the free text boxes beneath each contributing author's name to add specific details on the author's contribution.

More information is available in our guide to authors.
<https://www.embopress.org/page/journal/14602075/authorguide>

>> Section order should be corrected as follows: title page with complete author information, abstract, keywords, introduction, results, discussion, methods, data availability section, acknowledgements, disclosure and competing interests statement, references, main figure legends, tables, expanded figure legends. The data availability section needs to be placed before Acknowledgments.

>> Funding: please enter the following funding information in the list of funders in our online system: 'IRCM Foundation and FEDER funds/European Regional Development Fund (ERDF); 10.13039/501100011033'. Include the funding information into the Acknowledgments section.

>> Figure callouts: Please ensure that the Appendix Tables S2-S4 are called out in sequential order in the main manuscript text.

>>Appendix file with ToC: substitute the term 'Supplementary' with 'Appendix'.

>> Source data: the source data checklist needs to be provided separately as a Related Manuscript file and each source data figure folder should be provided separately.

>> The nomenclature of EV figure files and legends in the manuscript needs correction to 'Figure EV1', etc. .

>> Main tables (Table 1-3) need to be provided as separate, editable files.

>> Data availability section: please remove the referee token for the PRIDE and GEO datasets and make sure that these are made publicly accessible.

>> Recheck references for the bioRxiv entry Morin et al. (2024) and update the citation if in the meantime published as regular article.

>> Consider additional changes and comments from our production team as indicated below:

- DAS:

1. Please note that the specific URL for MSV000096690 dataset is not provided in the data availability statement.

2. Please note that reviewer access code for MSV000096690 dataset is provided in the manuscript.

>>> now added AD 5.8.25

3. Please note that reviewer access codes for GSE286238 dataset is not provided in the data availability statement.

>>> now added AD 5.8.25

- Figure legends:

1. Please note that the exact p values are not provided in the legends of figures 3A, 4D, F; 6D, 7C, EV1 B, EV3 B, D; EV4 D.

2. Please indicate the statistical test used for data analysis in the legends of figures 3B, 7A, B; EV1 E.

3. Please note that the box plots need to be defined in terms of minima, maxima, centre, bounds of box and whiskers, and percentile in the legends of figures 6B, C

4. Please note that information related to n is missing in the legends of figures 7A,

5. Please note that n=2 in figures EV2 A

Referee #1:

The authors have adequately addressed the criticism raised by this reviewer. The manuscript has improved in the revision process, and I support the publication of it.

Referee #3:

The authors have suitably adapted the manuscript with additional data to address points raised in the original version and have refined elements that were previously unclear.

I have no additional comments and thank the authors for their additional efforts.

The authors addressed the remaining editorial issues.

Dear Dr Rivera,

Thank you for submitting the revised version of your manuscript. We have now evaluated your amended study and concluded that the remaining minor concerns have been sufficiently addressed.

I am thus pleased to inform you that your manuscript has been accepted for publication in the EMBO Journal.

Related, I would like to hereby ask your consent on keeping the rebuttal figure included in this file.

On a different note, I would like to alert you that EMBO Press offers a format for a video-synopsis of work published with us, which essentially is a short, author-generated film explaining the core findings in hand drawings, and, as we believe, can be very useful to increase visibility of the work. Please see the following link for representative examples and their integration into the article web page:

<https://www.embopress.org/doi/full/10.1038/s44318-025-00417-0>

Finally, we have noted that the submitted version of your article is also posted on the preprint platform bioRxiv. We would appreciate if you could alert bioRxiv on the acceptance of this manuscript at The EMBO Journal in order to allow for an update of the entry status. Thank you in advance!

Best regards,

Daniel Klimmeck

Daniel Klimmeck, PhD
Senior Editor
The EMBO Journal
EMBO

Postfach 1022-40
Meyerhofstrasse 1
D-69117 Heidelberg
contact@embojournal.org
